# 14-3-3 binding maintains the Parkinson's associated kinase LRRK2 in an inactive state

Juliana A. Martinez Fiesco[1], Alexandra Beilina[2], Astrid Alvarez de la Cruz [1], Ning Li [1], Riley D. Metcalfe [1], Mark R. Cookson [2] & Ping Zhang [1] ✉

Leucine-rich repeat kinase 2 (LRRK2) is an essential regulator in cellular signaling and a major contributor to Parkinson's disease (PD) pathogenesis. 14-3-3 proteins are critical modulators of LRRK2 activity, yet the structural basis of their interaction has remained unclear. Here, we present the cryo-electron microscopy structure of the LRRK2:14-3-3$_2$ autoinhibitory complex, revealing how a 14-3-3 dimer stabilizes an autoinhibited LRRK2 monomer through dual-site anchoring. The dimer engages both phosphorylated S910/S935 sites and the COR-A/B subdomains within the Roc-COR GTPase region. This spatial configuration constrains LRR domain mobility, reinforces the inactive conformation, and likely impedes LRRK2 dimerization and oligomer formation. Structure-guided mutagenesis studies show that PD-associated mutations at the COR:14-3-3$_2$ interface and within the GTPase domain weaken 14-3-3 binding and impair its inhibitory effect on LRRK2 kinase activity. Furthermore, we demonstrate that type I LRRK2 kinase inhibitor, which stabilizes the kinase domain in its active conformation, reduces 14-3-3 binding and promotes dephosphorylation at pS910 and pS935. Together, these findings provide a structural basis for understanding how LRRK2 is maintained in an inactive state, elucidate the mechanistic role of 14-3-3 in LRRK2 regulation, inform the interpretation of PD biomarkers, and suggest therapeutic strategies aimed at enhancing LRRK2-14-3-3 interactions to treat PD and related disorders.

Mutations enhancing leucine-rich repeat kinase 2 (LRRK2) activity are a leading cause of familial Parkinson's disease (PD), and genetic variation at the same locus significantly contributes to lifetime risk of idiopathic PD[1–7]. LRRK2 is a large, 2527 residue multidomain protein containing both GTPase and kinase domains[8–14]. Its catalytic core comprises a Roco family GTPase domain, including a GTP-binding Ras of complex proteins (Roc) domain coupled with a C-terminal of Roc domain (COR, split into COR-A and COR-B subdomains), and a serine/threonine kinase domain. This core is flanked by N-terminal armadillo (ARM), ankyrin (ANK), and leucine-rich repeats (LRR) domains, and a C-terminal WD40 domain (Fig. 1a)[15–19]. LRRK2 plays crucial roles in the endolysosomal system, notably through the phosphorylation of specific Rab proteins[20–23]. Pathogenic mutations associated with PD,

located in the Roc, COR, and kinase domains (Supplementary Fig. 1a), typically increase kinase activity and/or decrease GTPase activity[1–7,24–33]. Elevated LRRK2 kinase activity has also been associated with an increased risk of cancer[34–38].

Although the precise molecular mechanisms driving LRRK2 kinase activation remain elusive, several factors, including phosphorylation[39,40], oligomerization[41–50], membrane association[51–56], and complex formation with regulatory proteins, likely contribute[57–60]. Recent studies suggest that LRRK2 is relatively inactive in cells unless triggered by damage to lysosomes, which then leads to the accumulation of phosphorylated Rab proteins on membranes[51–56]. Members of the 14-3-3 family are well-established LRRK2 interactors and have been proposed to contribute to LRRK2 stability[61,62].

[1]Kinase Complexes Section, Center for Structural Biology, Center for Cancer Research, National Cancer Institute, Frederick, MD, USA. [2]Cell Biology and Gene Expression Section, National Institute on Aging, National Institutes of Health, Bethesda, MD, USA. ✉e-mail: ping.zhang@nih.gov

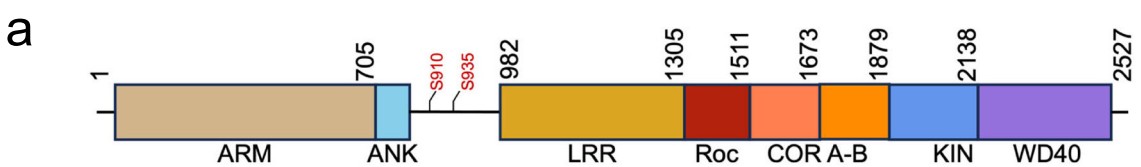

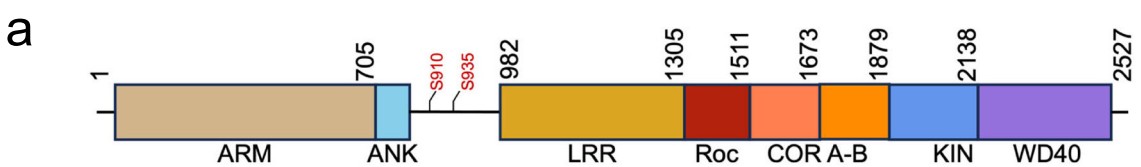

**Fig. 1 | Structure of the LRRK2:14-3-3₂ complex. a** Schematic representation of LRRK2 domain organization. Residues S910 and S935, which serve as 14-3-3 binding sites upon phosphorylation, are highlighted in red. **b** Cryo-EM density map at a resolution of 3.96 Å (left), with the corresponding structural model shown on the right. The model is colored according to the domain color code in (**a**) and shown in two different orientations for clarity.

14-3-3 proteins are ubiquitously expressed and highly abundant in cells. As regulatory proteins, they function as dimeric scaffolds[63,64] that modulate a broad spectrum of client proteins through various mechanisms[65–70]. Several PD-associated mutations such as R1441C/G/ H, Y1699C, and I2020T, exhibit reduced 14-3-3 interaction, which is associated with increased kinase activity[71–73]. Additionally, a study on PD rodent models and postmortem PD brain tissue reported reduced LRRK2 and 14-3-3 interactions, which were also associated with

increased kinase activity in idiopathic PD[74]. Collectively, these findings indicate that 14-3-3 binding modulates LRRK2 activity, often exerting an inhibitory effect.

14-3-3 proteins are phosphoserine/phosphothreonine-binding proteins[75]. Recent structural studies have revealed how 14-3-3 proteins interact with various kinases and scaffold proteins[76–79]. In LRRK2, the N-terminal region features a loop preceding the LRR domain that includes a cluster of potential 14-3-3 binding sites (Supplementary Fig. 1a)[61,80]. Additional potential 14-3-3 binding sites have also been suggested within the Roc domain and the C-terminus of the protein (Supplementary Fig. 1a)[81]. Despite extensive evidence of LRRK2 and 14-3-3 interactions, the exact molecular details, such as binding sites, stoichiometry, and the impact on LRRK2's oligomerization and kinase activity, remain unclear. This limits the mechanistic understanding of how 14-3-3 regulates LRRK2 under physiological conditions and how dysregulated interactions may contribute to PD pathogenesis.

Here, we report the cryo-EM structure of the full-length monomeric LRRK2 complexed with a 14-3-3 dimer, revealing how phosphorylation sites and GTPase subdomains engage to stabilize LRRK2 in an inactive state. Our findings provide a structural framework for understanding how 14-3-3 modulates LRRK2 kinase activity and lay the groundwork for future development of therapeutic strategies aimed at modulating this critical interaction.

## Results

### Formation and structural characterization of the LRRK2:14-3-3₂ complex

To study the interaction between LRRK2 and 14-3-3 proteins, we expressed and purified both proteins separately, using the monomeric form of LRRK2 and 14-3-3 gamma (γ), the most abundant 14-3-3 isoform in the brain[82] and the isoform with the highest affinity for LRRK2-derived peptides[81] (see "Methods" section, Supplementary Fig. 1). We successfully formed the LRRK2/14-3-3 complex under conditions optimized for ionic strength (Supplementary Fig. 2), enabling subsequent structural analysis. Mass spectrometry and mass photometry analyses confirmed the presence of both proteins in the complex (Supplementary Fig. 2).

Initial cryo-electron microscopy (cryo-EM) studies revealed 2D class averages and a 3D density map consistent with formation of the LRRK2/14-3-3 complex (Supplementary Fig. 3), although the density corresponding to 14-3-3 was poorly defined, with a substantial population of LRRK2 not bound to 14-3-3. To improve the complex homogeneity, we applied cross-linking with bis(sulfosuccinimidyl)suberate (BS3) prior to size-exclusion chromatography (Supplementary Fig. 4), which improved the density of 14-3-3 in the complex.

Subsequent 3D reconstruction of the cross-linked particles yielded a 3.96 Å map, with a density corresponding to a LRRK2 monomer and additional density matching a 14-3-3 dimer, confirming LRRK2/14-3-3 complex formation (Fig. 1b, Supplementary Figs. 5 and 6). The structural details, including visible α-helices and β-strands, were well resolved, consistent with a map reconstructed at this resolution (Supplementary Fig. 6). The complex demonstrated a 1:1 stoichiometry between a LRRK2 monomer and a 14-3-3 dimer (Fig. 1b), hereafter referred to as the LRRK2:14-3-3₂ complex.

Within the complex, LRRK2 adopted a conformation similar to the previously reported inactive LRRK2 monomer[45], with a Cα root mean square deviation (RMSD) of 0.4 Å (Supplementary Fig. 7a). The N-terminal region (residues 1-906), including the ARM and ANK domains, was not observed in the density map, suggesting that they are likely flexible. The elongated LRR domain covered the kinase domain, occluding the active site and thus preventing substrate access to the kinase domain. While the kinase domain was nucleotide free, density within the Roc domain suggested the presence of bound GDP. Additionally, the conformation of the switch I loop in the Roc domain corroborated this GDP-bound state (Supplementary Fig. 6e).

The 14-3-3 dimer, characterized by nine antiparallel α-helices per protomer and forming a cup-like shape with two client-binding grooves[70], is positioned adjacent to the catalytic core of LRRK2 and makes contacts with the COR domain. Local refinement for the Roc-COR:14-3-3₂ part modestly improved the resolution in this area to 3.87 Å (Supplementary Fig. 6b).

Interestingly, despite using purified recombinant 14-3-3γ for the LRRK2:14-3-3₂ complex formation, mass spectrometry revealed the presence of multiple 14-3-3 isoforms in the sample (Supplementary Fig. 2b), likely due to co-purification of endogenous isoforms with LRRK2 from mammalian cells. Given the high sequence conservation of the 14-3-3 client-interacting residues (Supplementary Fig. 7b), these observations suggest that LRRK2:14-3-3 complex forms and is stable in vivo. To investigate this hypothesis further, we co-expressed LRRK2 and 14-3-3γ in mammalian cells and successfully purified the complex, demonstrating in vivo complex formation (Supplementary Fig. 8).

### 14-3-3 dimer binds to pS910 and pS935 sites and the COR domain in LRRK2

14-3-3 proteins interact with client proteins through two main types of interactions: primary and secondary. Primary interactions involve phosphoserine/threonine-containing motifs binding to a conserved amphipathic groove in 14-3-3, formed by the α-3, α-5, α-7, and α-9 helices[83]. Secondary interactions, which are less common, involve larger interfaces between the globular domain of the client protein and additional surfaces on 14-3-3[83], contributing to complex specificity and stability. Identifying the primary 14-3-3 binding sites on LRRK2 has been challenging due to the absence of conventional binding motifs[84,85]. Sequence analysis of LRRK2 reveals no obvious canonical 14-3-3 binding sites (Supplementary Fig. 9a). Nevertheless, experimental data indicate that LRRK2 contains several highly phosphorylated serine residues between the ANK and LRR domains, particularly S910, S935, S955, and S973[41,61,86–88], raising questions about how 14-3-3 engages LRRK2 and what the stoichiometry of the complex LRRK2 and 14-3-3.

Our cryo-EM structure of the LRRK2:14-3-3₂ complex reveals that a 14-3-3 dimer engages LRRK2 at two distinct sets of interfaces: primary interactions occur at pS910 and pS935 motifs, while secondary interactions involve the COR domain (Fig. 2a–c). The cryo-EM density map showed substantial densities for the 14-3-3 binding motifs across residues 907–919 and 930–940, each bound to one protomer of the 14-3-3 dimer, with clear density for the phosphate groups at pS910 and pS935 (Fig. 2a). Their phosphorylation was supported by mass spectrometry data (Supplementary Fig. 9b). Notably, previously reported for its high flexibility in other LRRK2 structures[45,50,89], this region was significantly stabilized upon 14-3-3 binding. The phosphate groups at S910 and S935 were positioned to interact with the positively charged groove of 14-3-3, including the canonical R57, R132, and Y133 triad as well as K50 within the α-3 helix (Fig. 2b). Hydrophobic interactions, between LRRK2 residues 911–919 and 936–940 and the 14-3-3 binding groove, further stabilize the interaction. The intervening loop region (residues 920–929) connecting the two binding sites is dynamic with lower resolution, visible only at low contour levels, and could not be modeled (Supplementary Fig. 10a). Interestingly, this loop contains residue Q923, and a LRRK2 Q923H mutation has been reported in one Brazilian patient with positive family history of PD[90]. Similarly, the segment from residue 942 to 983 connecting these primary interaction motifs to the LRR was flexible and unresolved (Supplementary Fig. 10a). The observed primary interactions closely resemble those observed in the crystal structure of human 14-3-3 bound to LRRK2 phospho-peptides containing pS910 and pS935[91].

Mutation of S910 and S935 to alanine abolished 14-3-3 binding, as demonstrated in our co-immunoprecipitation (Co-IP) experiments using Flag-tagged LRRK2 and endogenous 14-3-3 (Fig. 2d). To obtain a more direct measurement of binding between LRRK2

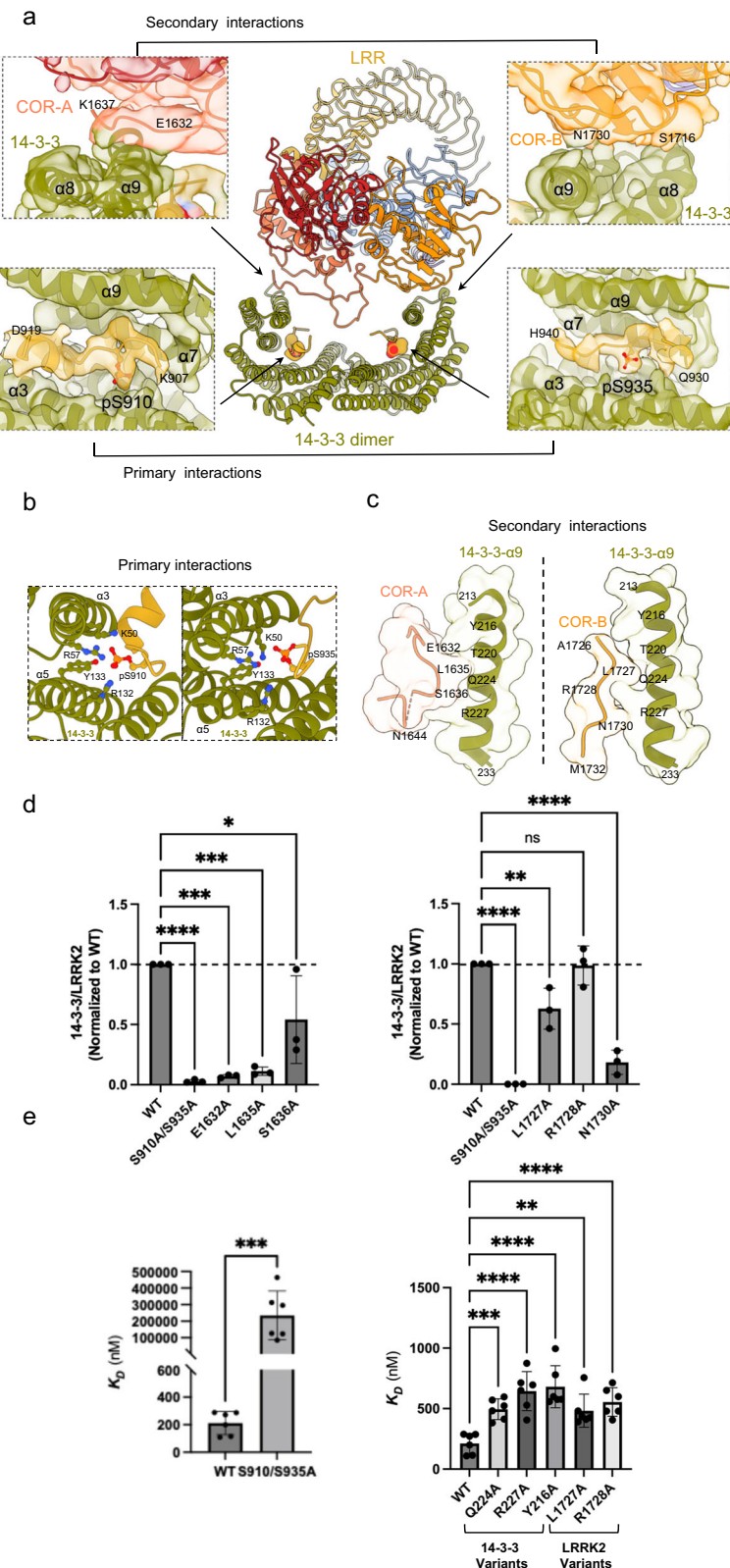

and 14-3-3, we performed microscale thermophoresis (MST) experiments. We found that the two proteins interact with a $K_D$ of 212 nM (Fig. 2e, Supplementary Fig. 11a). Moreover, mutations on S910 and S935 greatly reduce the affinity of the interaction (Fig. 2e, Supplementary Fig. 11b). This emphasized the critical role of these residues in forming the primary interactions with 14-3-3 and explained the dependency of the LRRK2:14-3-3 interaction on ionic

strength of the solution in vitro (Supplementary Fig. 2a). Substituting the 14-3-3 residues R132, R57, and K50 with alanine, located within the ligand-binding groove and known to interact with LRRK2 S910 and S935, as well as the triple mutant R57A/K50A/R132A resulted in impaired protein expression. This is likely because these residues are highly conserved and functionally critical; mutations at these positions may cause toxicity in the E. coli

**Fig. 2 | Detailed interactions and mutational effects at the LRRK2:14-3-3₂ binding interfaces. a** Overview of the LRRK2:14-3-3₂ contact regions in LRRK2:14-3-3₂ complex. Insets detail the primary and secondary interaction sites, supported by the corresponding cryo-EM densities. **b** Close-up view of the primary interactions, where LRRK2 phosphorylation sites pS910 and pS935 engage with the 14-3-3 substrate binding grooves. Assignment of this interaction is supported by integration of structural fitting, mass spectrometry, and biochemical validation. **c** Close-up view of the secondary interactions, showing LRRK2 COR-A and COR-B subdomain residues contacting the α−9 helices of the 14-3-3 dimer. **d** Quantitative analysis of LRRK2/14-3-3 interactions through Co-IP experiments of LRRK2 with endogenous 14-3-3, comparing WT LRRK2 with mutants at the secondary interface, as well as primary interface mutants (S910A/S935A). Data illustrate the impact of mutations on the interaction strength. Refer to Supplementary Fig. 16 for representative membrane images and source data for complete membrane images. Data are mean ± SEM ($n$ = 3 independent experiments), significance of difference was quantified using one-way Brown–Forsythe and Welch ANOVA test and reported with the exact p values in the source data file. **e** Binding affinity between (WT or mutant) LRRK2 and (WT or mutant) 14-3-3 proteins was determined by MST. Mutations at the primary (left) and secondary (right) binding sites were analyzed. Data are mean ± SEM ($n$ = 3 independent experiments), significance of difference was quantified using one-way Brown–Forsythe and Welch ANOVA test and reported with the exact p values in the source data file. Refer to Supplementary Fig. 11 for full binding curves.

expression system or result in protein misfolding and instability (Supplementary Fig. 12).

Affinity studies with synthetic peptides containing LRRK2 sequences previously suggested that residues within the Roc domain (S1444) and C-terminus (T2524) of LRRK2 may serve as additional 14-3-3 binding motifs[81]. However, in our structure, these sites are occluded by intramolecular interactions within LRRK2 (Supplementary Fig. 10b) and are not solvent-exposed, thus preventing direct interaction with 14-3-3. Furthermore, our mass spectrometry data detected no phosphorylation sites beyond residue 976 (Supplementary Fig. 9b), indicating that phosphorylation of LRRK2 in the LRRK2:14-3-3₂ complex is restricted to the N-terminal half of the protein; however, further studies with comprehensive sequence coverage will be needed to confirm this observation.

In addition to the canonical binding grooves, our LRRK2:14-3-3₂ structure revealed a distinct interface between the COR domain and the 14-3-3 dimer, constituting the secondary interactions between LRRK2 and 14-3-3. Residues from COR-A (1632–1644) and COR-B (1727–1732) interact with both α-9 helices in the 14-3-3 dimer (Fig. 2), burying 186 and 278 Å² of surface areas, respectively, predominantly through Van der Waals interactions. Specifically, LRRK2 COR-A residues E1632, L1635, and S1636 and COR-B residues L1727, R1728, and N1730 at the secondary interfaces make numerous contacts with 14-3-3 α-9 helices residues Y216, Q224, and R227 in the model (Fig. 2c).

To evaluate the contribution of these residues, we mutated these COR residues and performed Co-IP assays. We observed a reduction in binding of ~40–90%, with mutations E1632A, L1635A, and N1730A displaying the largest effect (Fig. 2d), demonstrating that while the segment containing the pS910/S935 establish the primary and strongest interaction with 14-3-3 dimer, the secondary interactions between 14-3-3 and COR-A and COR-B subdomains also contribute to the overall stability of the interaction between the two proteins. Furthermore, mutations on either side of the COR: 14-3-3₂ interface reduced the binding affinity, as measured by MST, leading to an approximately 2–3-fold increase in the $K_D$ values (Fig. 2e, Supplementary Fig. 11c), confirming the role of this secondary interface in stabilizing the complex.

In addition, as discussed in the next session and consistent with previous studies[61,62,71,72], LRRK2 activity was inhibited by 14-3-3. We further demonstrated that mutations that disrupt either primary or secondary interaction sites reduce this inhibition. While the secondary interface observed here, located outside the classic 14-3-3 cradle, is unusual, similar interfaces have been observed in other 14-3-3/client complexes (for example, BRAF/14-3-3 and Exoenzyme T/14-3-3 complexes[79,92]). The key residues involved in the primary interaction with LRRK2 are highly conserved among the different 14-3-3 isoforms (Supplementary Fig. 7b), and similarly, those involved in the secondary interface are conserved in all human 14-3-3 family members, indicating that a similar interface would be expected regardless of the isoform composition of the 14-3-3 dimer (Supplementary Fig. 7b).

## 14-3-3 binding to the COR domain interferes with LRRK2 dimerization and oligomerization

COR domains are well-documented dimerization modules in Roco proteins[17] and have been shown to facilitate the assembly of LRRK2 into various oligomeric forms under specific experimental conditions, including dimers, tetramers, and higher-order oligomers[45,50,93]. These oligomeric structures, if physiologically relevant, are significant for understanding both the function and pathological implications of LRRK2[94]. Our structural analysis revealed a partial overlap between the COR:14-3-3₂ interface and the LRRK2 COR:COR interface in the inactive homodimer. This interaction is mediated by the COR-B subdomains of each LRRK2 protomer and is critical for dimer formation[45] (Fig. 3a). Specifically, residues 1727–1730 in COR-B, which are essential for the LRRK2 homodimer interface, also engage with 14-3-3 in the LRRK2:14-3-3₂ complex (Fig. 3b). These findings suggest that LRRK2 homodimerization and 14-3-3 binding to the COR domain are mutually exclusive, as the COR domain acts as a secondary interaction site in the LRRK2:14-3-3₂ complex.

To investigate whether 14-3-3 can bind LRRK2 homodimers via the primary interaction sites at S910 and S935, which remain accessible in the dimer (Fig. 3a), we conducted a series of multi-angle light scattering (MALS) experiments using pre-formed LRRK2 dimers. These dimers were isolated by gel filtration and confirmed by mass photometry (Supplementary Fig. 1b), then incubated with increasing concentrations of recombinant 14-3-3γ (Fig. 3c, Supplementary Fig. 13). We observed no disruption of the LRRK2 dimer with increasing concentrations of 14-3-3γ. At higher concentrations, the measured molecular mass (643 KDa) was consistent with the binding of a single 14-3-3γ dimer to a single LRRK2 dimer. This suggests that, under the experimental conditions used, 14-3-3 binding occurs without disrupting the dimer and likely interacts only through the exposed S910 and S935 sites.

However, it remains to be determined whether the partial overlap between the COR:14-3-3₂ and COR:COR dimerization interfaces could destabilize the LRRK2 dimer under physiological conditions, where 14-3-3 is present at concentrations orders of magnitude higher than LRRK2. Conversely, formation of the LRRK2:14−3-3₂ complex, in which the COR interface is occluded, would likely prevent LRRK2 dimerization. In addition to interfering with dimer formation, 14-3-3 binding may also obstruct the assembly of higher-order molecular oligomers, such as the tetramer observed on cryo-EM grids. This tetramer consists of two inactive and two active LRRK2 monomers, and interface overlap may hinder its formation (Supplementary Fig. 10c). Furthermore, 14-3-3 binding could block LRRK2 filament formation along microtubules, which involves COR-COR interactions in a closed active conformation[93]. Together, these analyses suggest that 14-3-3 binding to the COR domain not only stabilizes the monomeric, inactive form of LRRK2 but may also prevent the formation of functional or pathological oligomeric assemblies.

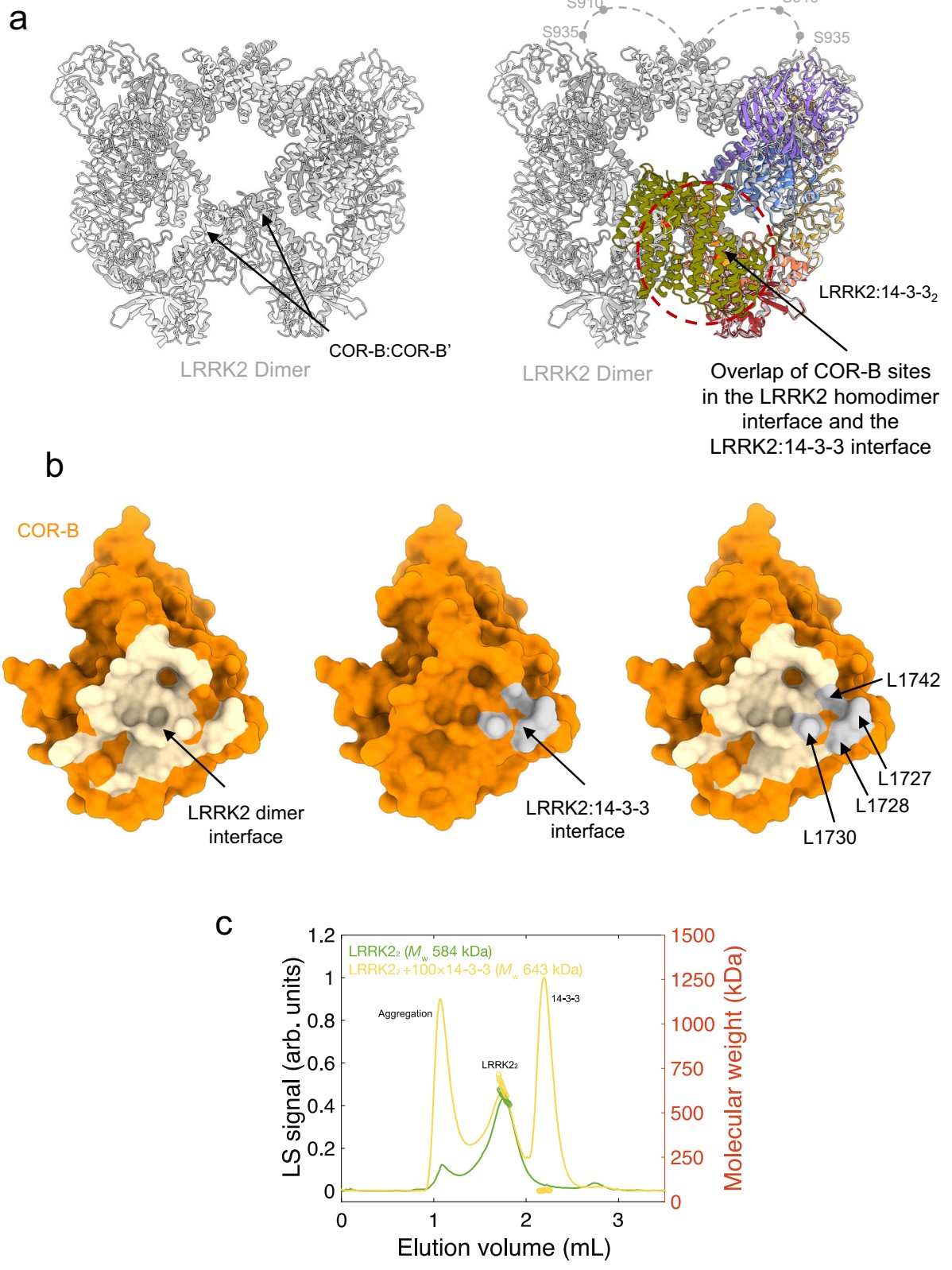

**Fig. 3 | Overlap of the LRRK2:14-3-3$_2$ and LRRK2 homodimer interfaces.**
**a** Structural comparison of the inactive LRRK2 homodimer (PDB: 7LHW, in gray, left) and the LRRK2:14-3-3$_2$ complex (colored as in Fig. 1), showing the overlapping interfaces mediated by the COR-B domain (right). Loops containing the S910 and S935 sites, flexible and unresolved in the dimer structure, are illustrated with dashed lines. **b** Surface representations of the COR-B domain. Left: residues involved in the LRRK2 dimer interface shaded in tan. Middle: residues involved in the LRRK2:14-3-3$_2$ interface shaded in gray. Right: Overlay of the two interfaces showing steric clash, suggesting mutual exclusivity between LRRK2 dimerization and 14-3-3 binding. **c** SEC-MALS chromatogram of the pre-formed LRRK2 dimer in the presence and absence of 14-3-3, indicating changes in molecular weight distribution.

## 14-3-3 binding inhibits LRRK2 kinase activity via dual-site anchoring

In our LRRK2:14-3-3$_2$ structure, the kinase domain adopts a canonical kinase-inactive conformation (Fig. 4a), consistent with previously reported inactive LRRK2 structures in the absence of 14-3-3 binding[45,50,89]. In this conformation, the Roc-COR domain is rotated away from the kinase domain, stabilizing it in an inactive state (Supplementary Fig. 7a). These observations prompted us to explore the role of 14-3-3 in modulating LRRK2 activity.

Our structure suggests that the 14-3-3 dimer does not induce the inactive LRRK2 conformation but rather maintains it by stabilizing the position of the LRR domain, a key regulatory element that adopts distinct configurations in the active and inactive states. In the inactive conformation (Figs. 1 and 4b), the LRR domain folds over the kinase domain, blocking substrate access and supporting the autoinhibited configuration. In contrast, in the active conformation, the LRR domain dramatically repositions away from the kinase domain, as observed in a Type I inhibitor-bound form[89] and a tetrameric assembly observed on cryo-EM grids[50]. In these active structures, both the LRR domain and the preceding loop (residues 907–982) are disordered and not resolved in the density map (Fig. 4b).

In the LRRK2:14-3-3$_2$ complex, the LRR domain and the 14-3-3 dimer are located on opposite sides of the Roc-COR-Kinase-WD40 region. The primary interaction between LRRK2 and 14-3-3 occurs at residues pS910 and pS935 within the loop immediately N-terminal to the LRR domain (residues 907–982), while the secondary interaction involves the COR-A and COR-B subdomains, which lie C-terminal to the LRR domain. Although the loop (residues 943–982) connecting the N-terminal phosphorylated motifs to the LRR domain is flexible and unresolved in the cryo-EM map, the dual-site anchoring of both N- and C-terminal flanking regions by the 14-3-3 dimer imposes a spatial constraint that limits the repositioning of the LRR domain relative to the kinase domain. Notably, this mechanism does not require direct contact between 14-3-3 and the LRR domain itself but rather relies on spatial constraints imposed by dual-site anchoring, reinforcing the kinase-inactive conformation.

To further explore this mechanism, we performed 3D variability analysis (3DVA) using *Cryosparc*[95]. This revealed that the LRR domain and the 14-3-3 dimer undergo coordinated movements relative to the Roc-COR-Kinase-WD40 core. These synchronized movements suggest that 14-3-3 flexibly tethers the regions flanking the LRR domain, restricting its conformational range and effectively maintaining LRRK2 in an autoinhibited state (Supplementary Fig. 14, Supplementary Movie 1).

We next validated the functional significance of this dual-site anchoring mechanism by assessing LRRK2 kinase activity in vitro. Using an in vitro kinase assay with Rab10 as a substrate, we found that the addition of 14-3-3 to LRRK2 inhibited Rab10 phosphorylation by up to ~50% in a concentration-dependent manner (Fig. 4c).

We next examined the impact of mutations at the LRRK2 and 14-3-3 interfaces on 14-3-3 mediated LRRK2 kinase activity inhibition (Fig. 4d, e). Mutants at the primary interaction residues S910/S935 and the secondary interaction residues L1727, R1728, and N1730 significantly reduced inhibition (~25% inhibition of Rab10 phosphorylation by 14-3-3 compared to a 50% inhibition with the wild-type (WT) protein, Fig. 4d). Similarly, mutations on the 14-3-3 interface residues Q224 and Y216 residues also reduced LRRK2 inhibition (~22% inhibition of Rab10 phosphorylation by 14-3-3 compared to a 50% inhibition with the WT protein, Fig. 4e), highlighting the importance of these interfaces in the 14-3-3 inhibitory effect. Additionally, we also observed that maintaining the hydrophobic nature of the secondary interface was crucial for the interaction. When residues such as R1728 and E1632 in the COR domain and R227 in 14-3-3 were mutated to more hydrophobic ones, the inhibition was reinforced. Moreover, mutations at E1632, L1635, and S1636 in LRRK2 were not completely tolerated, as

they resulted in lower expression yields or reduced thermostability (Supplementary Figs. 15 and 16).

## The active conformation of LRRK2 weakens 14-3-3 binding

To investigate how conformational changes associated with LRRK2 activation affect 14-3-3 binding, we modeled the interaction of 14-3-3 with the active conformation of LRRK2 by superimposing the active conformation onto the inactive LRRK2 in the LRRK2:14-3-3$_2$ complex (Fig. 5). In the active LRRK2 conformation, the Roc-COR domain rotates towards the kinase domain to stabilize its active state (Fig. 5a)[50]. In this configuration, the COR-B subdomain rotates relative to the COR-A, making it incompatible to engage 14-3-3 simultaneously with the COR-A (Fig. 5b). This orientation prevents the complete binding of 14-3-3 to the active form of LRRK2.

To test this model experimentally, we assessed the interaction between LRRK2 and 14-3-3 in the presence of a saturating concentration of a type I (MLi-2) or a type II (Rebastinib) kinase inhibitor using MST. We observed that the binding affinity between LRRK2 and 14-3-3 was decreased by 2-fold in the presence of the type I inhibitor (MLi-2) (Fig. 5c, Supplementary Fig. 17). In contrast, the binding affinity was unaltered in the presence of the Type II inhibitor (Rebastinib). These findings support the model that conformational changes during LRRK2 activation impair 14-3-3 binding by disrupting the COR:14-3-3$_2$ interfaces.

This result aligns with prior observations showing that treatment with type I inhibitors leads to loss of LRRK2 phosphorylation at S910/S935[61]. These inhibitors are known to stabilize LRRK2 in an active conformation[61], albeit not able to phosphorylate downstream substrates due to the presence of ATP-competitive inhibition. However, the precise mechanisms by which these inhibitors alter LRRK2 phosphorylation and influence the stabilization of the active conformation were not well understood. Our findings suggest that, in vivo, destabilization of the inactive conformation and formation of the active conformation disrupts the LRRK2:14-3-3$_2$ complex, leading to exposure of the S910/S935 phosphorylation sites for phosphatase activity. To test this mechanism in cells, we measured the LRRK2 pS910 and/or pS935 levels after treatment with MLi-2 or Rebastinib, as well as in cells expressing LRRK2 mutants that destabilize the COR:14-3-3$_2$ interface. Consistent with our model, we observed a ~19-fold reduction in LRRK2 pS910 and pS935 levels in cells treated with the type I inhibitor MLi-2, accompanied by a ~7-fold decrease in co-immunoprecipitated 14-3-3 levels (Supplementary Fig. 18a). In contrast, treatment with the type II inhibitor Rebastinib had no significant effect (Supplementary Fig. 18a). Likewise, destabilization of the COR:14-3-3$_2$ interface through L1727A, R1728A, N1730A, E1632A, and L1635A mutations led to a ~2- to 5-fold reduction in pS935 levels (Supplementary Fig. 18b), which was accompanied by a 2- to 13-fold decrease in co-immunoprecipitated 14-3-3 levels (Fig. 2d). These findings support a model in which conformational shifts associated with LRRK2 activation destabilize the LRRK2:14-3-3$_2$ complex, facilitating dephosphorylation at S910/S935.

Together, these results provide a mechanistic explanation for the paradoxical effects of Type I inhibitors: while they inhibit LRRK2 kinase activity directly, they induce an active LRRK2 conformation that disrupts 14-3-3 interactions, leading to S910/S935 dephosphorylation. More broadly, our findings shed light on the importance of conformational transitions in controlling LRRK2 regulation and demonstrate that dual-site 14-3-3 anchoring is essential for maintaining the inactive state of the kinase.

## Pathological mutations that disrupt complex stability and LRRK2 inhibition by 14-3-3

Several prominent PD mutations, such as Y1699C and R1441C/H/G, located in the Roc-COR GTPase domain of LRRK2, exhibit elevated kinase activity[23,94]. This region not only governs GTPase and kinase function but also mediates oligomerization and regulatory interactions. Our findings demonstrate that the COR domain is critical not

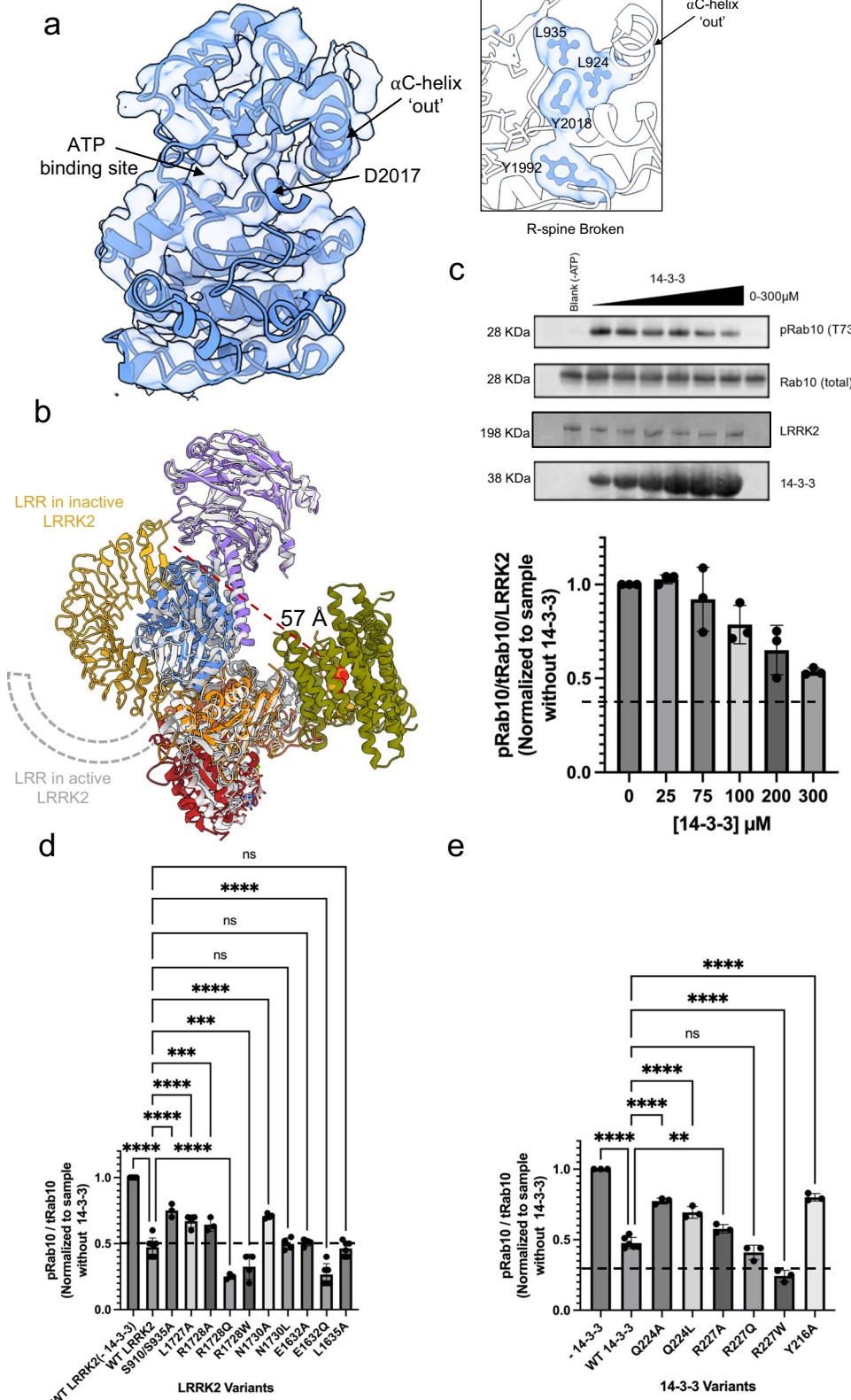

**Fig. 4 | 14-3-3 binding maintains LRRK2 in an inactive conformation and inhibits its kinase activity. a** Structural representation of the kinase domain in the LRRK2:14-3-3₂ complex, showing an inactive conformation characterized by an outward αC helix and a broken regulatory (R-) spine (inset). **b** Overlay of the LRRK2:14-3-3₂ complex (colored as in Fig. 1) with the active LRRK2 monomer from a LRRK2 tetramer (PDB: 8FO9, in gray), highlighting the conformational differences in the LRR domain, depicted as a cartoon. **c** In vitro kinase assay showing inhibition of LRRK2 activity by increasing concentrations of 14-3-3, measured by Rab10

phosphorylation levels (pRab10) in western blots, normalized to LRRK2 protein levels. Effects of interface mutations on kinase activity inhibition by 14-3-3, with results for LRRK2 mutations shown in (**d**) and 14-3-3 mutations in (**e**). Data in (**c**–**e**) are mean ± SEM (n = 3 independent experiments), with representative blots for inhibition assays shown in (**c**). See source data for membrane images and for significance of difference with the one-way Brown–Forsythe and Welch ANOVA test with the exact *p* values when applicable.

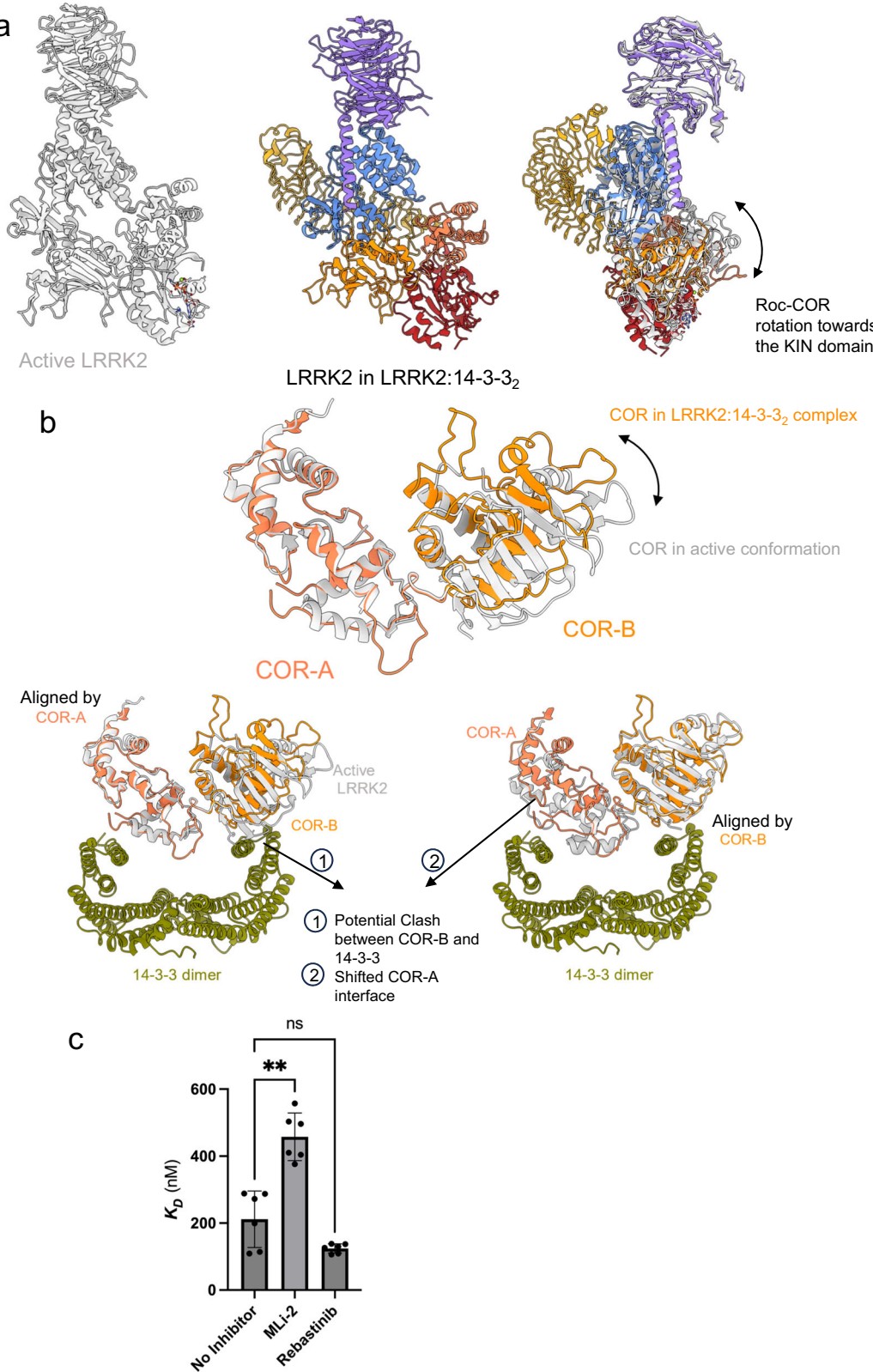

**Fig. 5 | Structural comparison of the LRRK2:14-3-3₂ complex with active LRRK2.** **a** Side by side comparison of the LRRK2:14-3-3₂ complex (colored as in Fig. 1) and the active conformation of LRRK2 derived from a tetrameric structure (PDB: 8FO9, in gray), highlighting the Roc-COR domain rotation (indicated by an arrow). 14-3-3 proteins are omitted for clarity. **b** Superimposition of the COR domain from the LRRK2:14-3-3₂ complex (shown in coral) with the COR domain from the active LRRK2 (shown in gray). Alignments were performed by the COR-A domain (top and bottom left) and by the COR-B domain (bottom right), to illustrate the structural shifts and effects. **c** Binding affinity measurements of LRRK2 and 14−3-3γ in the presence of kinase inhibitors, as determined by MST in vitro. Data in (**c**) are mean ± SEM (*n* = 3 independent experiments), significance of difference was quantified using one-way Brown−Forsythe and Welch ANOVA test and reported with the exact *p* values in the source data file. Refer to Supplementary Fig. 17 for full binding curves.

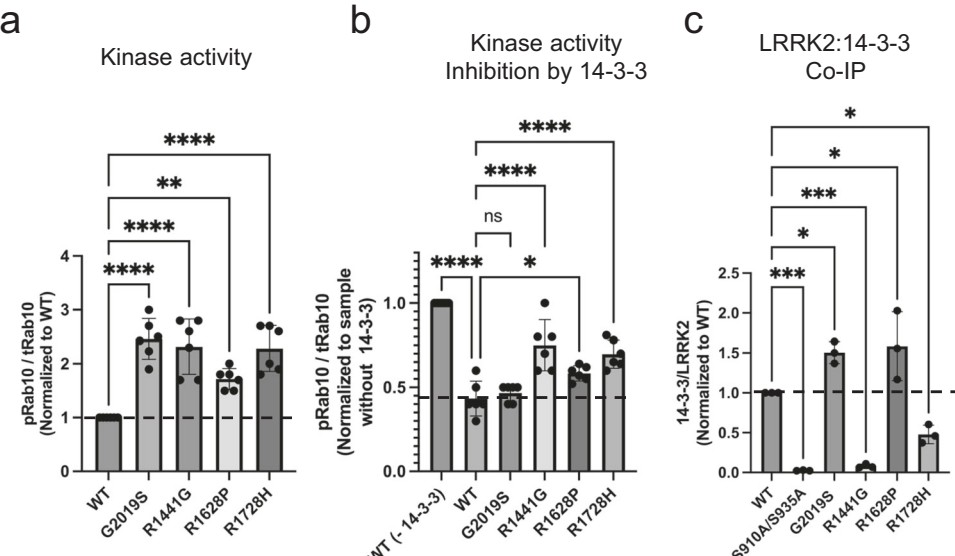

**Fig. 6 | Impact of PD-associated mutations in Kinase and GTPase domains on LRRK2:14-3-3₂ interaction. a** In vitro LRRK2 kinase activity assay comparing wild-type (WT) and various PD-related hyperactive mutants in the kinase and GTPase domains, showing increased Rab10 phosphorylation levels indicative of enhanced kinase activity. **b** Comparative analysis of 14-3-3 inhibition of kinase activity across WT and PD mutants, illustrating that mutations differentially impair LRRK2 regulation by 14-3-3. **c** Co-IP assays quantifying cellular interaction between LRRK2

WT/PD-related mutations and endogenous 14-3-3 in cells, revealing altered binding affinities caused by specific mutations. LRRK2 kinase activity and co-IP data (**a**–**c**) are mean ± SEM (n = 3 independent experiments). Refer to Supplementary Fig. 16 and source data for membrane images and for significance of difference with the one-way Brown–Forsythe and Welch ANOVA test with the exact p values when applicable.

only for oligomerization but also for interaction with 14-3-3 proteins and that this can be disrupted by PD-related mutations.

We examined the effects of PD-linked mutations R1628P and R1728H, located at or near the COR-A/14-3-3 and COR-B/14-3-3 interfaces, respectively. Both mutations led to a ~ two-fold increase in kinase activity in vitro (Fig. 6a), consistent with earlier in vivo reports[96–99]. Additionally, these mutations showed a ~20% reduction in LRRK2 inhibition by 14-3-3 compared to WT (Fig. 6b). Co-IP experiments showed that R1728H resulted in a ~50% decrease in binding with endogenous 14-3-3 relative to WT (Fig. 6c), highlighting the critical impact of these interactions on LRRK2 kinase activity and PD pathogenesis. Additionally, PD-related mutations at the Roc-COR-Kinase-WD40 region, such as I2020T, L1795F, Y1699C, R1441C/H/G, A1442P, and G2385R, though positioned away from the direct LRRK2:14-3-3₂ interface, have been reported to disrupt 14-3-3 interactions[23,94]. These residues are not solvent-exposed and are implicated in various intramolecular interactions within the LRRK2 inactive conformation (Supplementary Fig. 10b). To investigate whether these mutations destabilize 14-3-3 interaction indirectly, we focused on two: R1441G (in the Roc domain) and G2019S (in the kinase domain). Both mutations exhibited approximately a 2.5-fold increase in kinase activity; strikingly, however, only the R1441G mutation led to a loss of interaction with 14-3-3 (Fig. 6c) and a roughly 30% reduction in 14-3-3 mediated inhibition of LRRK2 (Fig. 6b), indicating that R1441G disrupts the Roc-COR dynamics in a way that impairs 14-3-3 binding.

We examined the pS910 and pS935 levels in cells expressing LRRK2 pathogenic mutants. R1441G mutant caused a dramatic reduction in pS910 (~10-fold) and pS935 (~15-fold) levels, while the G2019S mutant showed only a modest reduction in pS910 (~1.5-fold) and no change in pS935 levels. Similarly, R1628P and R1728H mutants showed a ~3.0- and 1.6-fold reduction in pS935 levels, respectively (Supplementary Fig.18c). These results suggest that destabilization of the COR:14-3-3₂ interface leads to increased exposure of the pS910/S935 sites, making them susceptible to dephosphorylation. These findings further support the notion that loss of 14-3-3 binding correlates with dephosphorylation at key regulatory sites and with

increased kinase activity. Importantly, our results indicate that not all the PD-activating mutations have the same mechanism of hyper-activation. G2019S might enhance kinase activities through kinase kinetics[45,100] but does not disrupt the inactive conformation or 14-3-3 binding (Fig. 6c). By contrast, R1441G appears to destabilize the Roc-COR regulatory module, indirectly weakening 14-3-3 interaction and promoting LRRK2 activation. This mechanism may be conceptually similar to the effects of Type I inhibitors, though this hypothesis merits further studies.

## Discussion

Enhanced pathological LRRK2 kinase activity is critically implicated in PD pathogenesis, highlighting the physiological need for the precise regulation of kinase activity[23,94]. Here, we elucidated the structural and mechanistic basis of the interactions between LRRK2 and 14-3-3 proteins. Our findings reveal that 14-3-3 acts as both an inhibitor and regulator of LRRK2, crucial for modulating its activity under both physiological and pathological conditions.

Our structural analysis demonstrated extensive interactions between 14-3-3 proteins with key phosphorylation sites (pS910 and pS935) and the COR-A and COR-B subdomains of LRRK2 within the LRRK2:14-3-3₂ complex. These interactions stabilize LRRK2 in an inactive monomeric conformation, restricting the mobility of the LRR region and preventing substrate access. In contrast, in the active conformation of LRRK2, the LRR domain is repositioned away to expose the kinase domain, facilitating substrate access and highlighting potential targets for interventions to maintain LRRK2 in its inactive state. Interestingly, the positioning of the LRR domain distinguishes LRRK2 from its homolog LRRK1[101]. LRRK1 lacks the N-terminus phosphoserine binding motifs necessary for the primary interaction with 14-3-3[101,102] and differs at the secondary interface, explaining why LRRK1 is not regulated by 14-3-3 in the same way as LRRK2 (Supplementary Fig. 10d). This underscores a fundamental divergence in the regulatory mechanisms of LRRK1 and LRRK2.

Our mutagenesis studies, both in vitro and in cells (Fig. 6), revealed that pathological mutations at the COR:14-3-3₂ interfaces and

other areas within the Roc-COR GTPase domain substantially reduce 14-3-3 binding and subsequently enhance LRRK2 kinase activity. These findings emphasize the crucial role of 14-3-3 interactions in controlling LRRK2 activity. Our results also indicate that 14-3-3 binding maintains LRRK2 in its inactive conformation and prevents the formation of LRRK2 dimers and higher oligomers. Given the relatively high endogenous expression levels of 14-3-3 compared to LRRK2, the LRRK2:14-3-3$_2$ complex likely represents the dormant state of LRRK2 within cells. Furthermore, our analyses indicate that the COR conformation in the active state of LRRK2 prevents simultaneous engagement with both COR-A and COR-B, thereby blocking stable complex formation with active LRRK2. This model is supported by the observed reduction in LRRK2 phosphorylation at S910/S935 in the presence of type I inhibitors[61], which favor the active conformation of LRRK2. These inhibitors likely disrupt both secondary and primary interactions with 14-3-3 proteins, potentially altering LRRK2's functional state and impacting its regulatory mechanisms. We further tested this hypothesis by evaluating the effects of type I (Mli-2) and type II (Rebastinib) inhibitors on LRRK2 and 14-3-3 binding in vitro using MST (Fig. 5, Supplementary Fig. 17). In addition, we used cell-based assays to examine inhibitor-induced changes in LRRK2 pS910 and pS935 phosphorylation (Supplementary Fig. 18a). We show that the Type I LRRK2 kinase inhibitor, which stabilizes the active conformation of the kinase domain, reduces 14-3-3 binding and promotes dephosphorylation at pS910 and pS935, whereas the Type II inhibitor does not have such effects. These findings support our model in which conformational shifts associated with LRRK2 activation destabilize the LRRK2:14-3-3$_2$ complex, thereby facilitating dephosphorylation at S910/S935. Given these complexities, critical questions about LRRK2's functional states arise: How is LRRK2 converted from its 14-3-3 bound inactive form to an active form? And what constitutes LRRK2's physiological active form? It is possible that the monomeric, active form of LRRK2, which is free from 14-3-3 binding but may interact with other cofactors or be favored under specific cellular conditions, represents its most active state. Further experimental exploration is necessary to fully understand LRRK2 dynamics and activation mechanisms.

Our results also have significant implications for PD biomarkers and therapeutics. Phosphorylation levels of LRRK2 at S910/S935 and LRRK2 substrates Rab8/10 are widely used as biomarkers in clinical trials for monitoring PD progression and therapeutic responses[14,103–107]. Previous studies have shown that dephosphorylation at S910/S935 and elevated Rab8/10 phosphorylation occur in both familial and idiopathic PD[40]. Here, we provide mechanistic insights into how phosphorylation at these sites is important for LRRK2's interaction with 14-3-3 proteins to maintain its inactive state, and how the active conformation of LRRK2 can promote the exposure of these sites for dephosphorylation, leading to elevated Rab substrate phosphorylation levels.

We also show how Type I inhibitors reduce S910/S935 phosphorylation levels, a readout commonly used to assess target engagement and therapeutic response in trials of LRRK2 inhibitors. These inhibitors stabilize LRRK2 in an active conformation that is ATP-competitively inhibited, thus impairing its 14-3-3 binding and making pS910/pS935 sites more susceptible to dephosphorylation. Based on our findings, we proposed that the use of S910/S935 phosphorylation levels as biomarkers may need to be tailored to specific LRRK2 mutations that promote the active conformation of LRRK2 and impair 14-3-3 binding. This approach may also be relevant to idiopathic PD, which has been associated with reduced LRRK/14-3-3 interactions[74]. While it remains unclear whether LRRK2 adopts an active conformation in idiopathic PD, the observed decrease in 14-3-3 binding and increased susceptibility of pS910/pS935 to dephosphorylation in patient samples suggest that these phosphorylation sites may serve as useful biomarkers for stratifying LRRK2 regulatory states and identifying patients with similar LRRK2 regulatory dysfunctions. These markers may also help evaluate therapeutic responses to inhibitors that trap

LRRK2 in an active conformation. Our findings support the continued use of these phosphorylation markers in clinical trials and provide structural and mechanistic insights into how these biomarkers reflect LRRK2's conformational state and interaction with 14-3-3. This understanding strengthens the rationale for using these biomarkers and may guide more precise patient stratification in clinical settings.

Based on our findings, we propose that stabilizing the interaction between LRRK2 and 14-3-3 proteins using 'glue' molecules[108–110] could be a promising therapeutic strategy. Such molecules could enhance the interaction between LRRK2 and 14-3-3 proteins and suppress the pathological hyperactivity associated with LRRK2 mutations. While this concept remains to be experimentally validated, it represents a promising direction for future drug development in PD and related neurodegenerative disorders.

In summary, our study provides a structural framework for understanding how 14-3-3 proteins regulate LRRK2 and how PD-associated mutations impact their interactions. These findings offer valuable insights into the molecular mechanisms underlying LRRK2 regulation and lay a foundation for future therapeutic strategies targeting this pathway. Our work also informs the interpretation of PD biomarkers and supports further exploration of 14-3-3-based regulatory mechanisms in disease contexts.

## Methods
### Reagents and resources
Resources used in this study are listed in Supplementary Table 1. Plasmids and cell lines are available for use upon reasonable request to the corresponding author.

### Cell lines and culture conditions
Expi293F™ cells (Thermo Fisher Scientific, cat. A14527) were cultured in Expi293™ or FreeStyle™ 293 expression media at 37 °C with 8% $CO_2$.

### Expression and purification of LRRK2
Full-length human LRRK2 (Uniprot Q5S007 with R85H mutation), tagged with an N-terminus 3X-FLAG and a rhinovirus 3C protease cleavage site, was codon-optimized for *Homo Sapiens* and synthesized by GenScript. The gene was subcloned into the pEG BacMam vector for mammalian cell protein expression[111]. Bacmid was generated using the Bac-to-Bac system from Invitrogen and transfected into Sf9 cells for baculovirus generation. Expi293F cells were seeded at $2.0 \times 10^6$ cells/mL in four liters of FreeStyle™ 293 expression medium and infected with high-titer baculovirus. After ~12 h of incubation at 37 °C with shaking, 10 mM sodium butyrate was added, and the temperature was reduced to 30 °C. Cells were collected by centrifugation after 72 h of incubation, and cell pellets were flash frozen for later purification or resuspended in resuspension buffer (20 mM Tris pH 8.3, 10 mM $CaCl_2$, 5 mM $MgCl_2$, 100 mM $NH4Cl$, 100 mM NaCl, 10 mM β-glycerophosphate and 1 mM sodium vanadate, 50 mM L-Arg, 50 mM L-Glu, 0.0008% Tween-80, 10% glycerol).

All subsequent purification steps were carried out at 4 °C. Cells were lysed using sonication, and the clarified lysate was incubated with FLAG-M2 affinity resin (Sigma) for 2 h with rotation. The resin was washed once with resuspension buffer, twice with wash buffer (20 mM Tris pH 7.4, 500 mM NaCl, 5 mM MgCl2, 0.0008% Tween-80), and then once with gel filtration buffer (20 mM HEPES pH 8.3, 150 mM NaCl, 5 mM MgCl2, 0.0008% Tween-80). For elution, the column was washed three times with gel filtration buffer supplemented with 150 µg/mL 3X-FLAG peptide. The eluted protein was concentrated to ~500 µL and injected onto a Superose 6 Increase 10/300 column (Cytiva) equilibrated in gel filtration buffer. Peak fractions corresponding to LRRK2 monomer were collected for complex formation with 14-3-3 protein. LRRK2 mutants were obtained from GenScript and purified as described for WT LRRK2. Protein concentrations were determined using extinction coefficients calculated from the protein sequence.

For kinase assays, WT and mutant LRRK2 proteins were expressed in small-scale (50-200 mL) cell cultures, with expression and purification carried out as described above, with minor modifications. Lysis was carried out by incubating the cells in resuspension buffer supplemented with 1% Tween-80 at 4 °C for 1 h. After clarification, the lysate was incubated with 50 µL FLAG-M2 affinity resin for 2 h, followed by washes as described above. A single protein elution was done with gel filtration buffer supplemented with 150 µg/mL 3X-FLAG peptide. The eluted protein was filtered through a 0.22 µm Ultrafree-GV centrifugal filter (Sigma cat. UFC30GVNB) to remove residual resin prior to kinase and inhibition assays.

## Expression and purification of 14-3-3γ

N-terminal His-tagged 14-3-3γ (Uniprot P61981), with a tobacco etch virus (TEV) protease cleavage site and codon-optimized for Escherichia coli, was synthesized by Genscript and subcloned into a pET21b vector. This construct was transformed into E. coli BL21*(DE3) cells for protein expression. The starter cultures were grown in LB medium with 0.1 mg/ml ampicillin overnight at 37 °C and then diluted 1:100 into the same medium. The cultures were grown at 37 °C until the OD600 reached ~0.5–0.6. Protein expression was induced by adding 1 mM isopropyl β-D-thiogalactoside (IPTG), and the cultures were subsequently incubated for 18–20 h at 16 °C. Cells were harvested by centrifugation, resuspended in lysis buffer (25 mM Tris-HCl pH 7.4, 150 mM NaCl, 20 mM Imidazole, 1 mM TCEP, and protease inhibitors from Roche), and lysed by several passes through a microfluidizer (LM20 Microfluidizer, Microfluidics Corp). All subsequent purification steps described below were carried out at 4 °C. The lysates were centrifuged, and the collected supernatants were incubated with Ni-nitrilotriacetic (Ni-NTA) agarose beads for 0.5 h. The beads were washed with five column volumes of lysis buffer. The protein was eluted with elution buffer (25 mM Tris-HCl pH 7.4, 150 mM NaCl, 500 mM Imidazole, 1 mM TCEP) in a gradient over twenty column volumes. Fractions containing 14-3-3 were pooled, concentrated to ~5 mL, and injected onto a Superdex 75 16/160 column (GE Healthcare), which was equilibrated in gel filtration buffer. Fractions containing purified 14-3-3 were pooled, concentrated to ~10 mg/mL, and stored at −80 °C for long-term use. Mutants of 14-3-3 were generated using the NEB Q5 Site-Directed Mutagenesis Kit (cat. E0554). Protein concentrations were determined by UV absorbance at 280 nm using extinction coefficients calculated from the protein sequence.

## Expression and purification of Rab10

N-terminal His-SUMO-tagged Rab10 (Uniprot P61026), codon-optimized for expression in *E. coli*, was synthesized by Genscript and subcloned into a pET21b vector. The constructs were transformed into *E. coli* BL21 cells. Starter cultures were grown in LB medium supplemented with 0.1 mg/ml Ampicillin at 37 °C overnight and then diluted 1:100 into the same medium. The cultures were grown at 37 °C until the cell density reached ~0.5–0.6 $OD_{600}$ and induced with 1 mM isopropyl β-D-thiogalactoside (IPTG) and continued at 19 °C for 18–20 h. Cells were harvested by centrifugation, resuspended in lysis buffer (25 mM Tris 8.0, 500 mM NaCl, 1 mM MgCl2, 1 mM TCEP, and protease inhibitors from Roche), and lysed by sonication. All subsequent purification steps described below were carried out at 4 °C. The lysates were centrifuged, and the collected supernatants were incubated with Ni-nitrilotriacetic (Ni-NTA) agarose beads for 0.5 h.

The beads were washed with 60 column volumes of lysis buffer, followed by 20 column volumes each of lysis buffer supplemented with 10 mM and 25 mM imidazole, respectively. The protein was eluted with elution buffer (25 mM Tris-HCl pH 8.0, 100 mM NaCl, 500 mM Imidazole, 1 mM TCEP) over a gradient of twenty column volumes. Eluates were spin dialyzed into the lysis buffer, after which NP-40 was added to a final concentration of 0.1% and subjected to UlP1 (an engineered SUMO protease) digestion for 1 h at 25 °C at a molar ratio of 1:200 (protein:enzyme) to cleave the His-SUMO tag. The cleaved tag and the protease were then removed by a second round of Ni-NTA purification. The Rab10 fractions were pooled, concentrated to ~5 mL, and injected onto a Superdex 75 16/160 column (GE Healthcare) equilibrated in gel filtration buffer. Fractions containing purified Rab10 were taken, pooled, concentrated to ~10 mg/mL, and stored at −80 °C for long-term use. Protein concentrations were determined by UV absorbance at 280 nm using extinction coefficients calculated from the protein sequence.

## LRRK2/14-3-3 complex formation

Purified LRRK2 monomer fraction was diluted in gel filtration (GF) buffer with no NaCl to reduce NaCl concentration to 50 mM and then incubated with at least a 60-fold molar excess of 14-3-3 protein on ice for 15 min to promote complex formation. For experiments requiring cross-linking, the protein mixture was first incubated for 10 min at room temperature, followed by the addition of 1 mM bis(sulfosuccinimidyl)suberate (BS3). After a further 15-min incubation at room temperature, the reaction was quenched with 50 mM Tris, pH 7.4. The cross-linked LRRK2/14-3-3 complex was concentrated to ~500 µL and injected onto a Superose 6 Increase 10/300 column (Cytiva), which was equilibrated in gel filtration buffer. Fractions from the peak corresponding to the cross-linked LRRK2/14-3-3 complex were used for cryo-EM grid preparation.

## Mass photometry

Samples containing LRRK2 or LRRK2/14-3-3 complexes were diluted to concentrations ranging from 20 to 50 nM in detergent-free gel filtration buffer for mass photometry measurements using a OneMP mass photometer (Refeyn). Movies were collected for 6000 frames over 60 s in regular view mode. Mass determination was conducted using the DiscoverMP software, with calibration performed using a mixture of beta amylase (Sigma, cat. A8781) and thyroglobulin (Sigma, cat. T9145).

## Multi-angle static light scattering (MALS)

SEC-MALS data were collected using a Shimadzu LC-20AD HPLC, coupled to a Shimadzu SPD-20A UV detector, a Wyatt Dawn MALS detector, and a Wyatt Optilab refractive index detector. Data were collected following in-line fractionation with a Superose 6 Increase 15/150 column (GE Healthcare), pre-equilibrated in gel filtration buffer, operated at a flow rate of 0.3 mL/min. For each run, 50 µL of the dimeric LRRK2 at 130 nM was applied to the column for analysis in the absence and presence of increasing concentrations of 14-3-3γ, ranging from 1 to 100 times that of LRRK2. Data were processed using ASTRA software v. 8.0.2.5 (Wyatt). The detector response was normalized using monomeric BSA (Thermo Fisher, cat. 23209). Protein concentration was determined using differential refractive index, using a dn/dc value of 0.185 mL/g.

## LRRK2 kinase activity assays and inhibition assays with 14-3-3

Kinase reaction mixtures consisted of 100 nM LRRK2 and 3 µM Rab10, in 20 mM HEPES buffer pH 7.4, 50 mM NaCl, 5 mM $MgCl_2$, and 0.0008% Tween-80. Reactions were carried out at 30 °C for 30 min in a thermomixer (Eppendorf) with shaking (300 rpm). Reactions were initiated by adding 5 mM ATP and terminated by the addition of LDS-NuPAGE loading buffer (Thermo Fisher). The samples were boiled at 95 °C for 10 min and stored at −80 °C if not processed immediately in immunoblot analysis. Kinase assays were performed using at least three independent protein preparations, each in duplicate. Samples were resolved on 4–12% bis-tris gels and wet-transferred to a 0.4 µm PVDF membrane. Membrane blocking was done with 5% bovine serum albumin (BSA) in TBS-T (20 mM Tris, 150 mM NaCl pH 7.4, 1% Tween-20) for 30 min at room temperature before probing with anti-phospho Rab10 T73 primary antibody (Abcam, cat. ab241060, dilution 1:500),

anti-Rab10 primary antibody (Abcam cat. ab237703, dilution 1:1000), and anti-LRRK2 primary antibody (Abcam cat. ab133474, dilution 1:25,000). Membranes were washed with TBS-T three times and then incubated with goat anti-rabbit and anti-mouse IR-fluorescent secondary antibodies (Li-Cor, cat. 926-3221, Li-Cor, cat. 926-68072) for 1 h at room temperature. Following four washes, membranes were imaged using a Typhoon scanner (GE Healthcare, software v. 1.1.0.7). Blots were quantified using ImageStudio Lite software (v.5.2.5) to determine the pRab10/Rab10 ratio, normalized to WT controls on the same membrane. Statistical significance was quantified using a one-way Brown–Forsythe and Welch ANOVA test in Prism (v.10.2.0), with pairwise comparisons via unpaired t-tests with Welch's correction.

For inhibition assays with 14-3-3 proteins, kinase reactions were carried out as described above with an additional incubation of the reaction mixture with 14-3-3 on ice for 15 min prior to ATP addition. The dependency of LRRK2 inhibition on 14-3-3 concentration was evaluated in a range from 0 to 300 μM. The effects of mutations in LRRK2 or 14-3-3 were tested using 14-3-3 at 300 μM. Additionally, following Rab10 phosphorylation probing, blots were further analyzed with anti-14-3-3γ antibody (Abcam, cat. ab137048, dilution 1:1000).

### Cryo-EM grid preparation, data acquisition, and processing

After gel filtration, the LRRK2/14-3-3 complexes were concentrated to ~0.15–0.20 mg/mL using a 100 K pore size Pall Microsep™ advance centrifugal device. Quantifoil Au R1.2/1.3 holey carbon grids, 300 mesh, were glow-discharged for 30 s at 25 mA on both sides. 1.5 μL of protein solution was applied to each side of the grids. Grids were vitrified using a Leica EM GP2 plunge freezer with a blotting time of 1–3 s. Cryo-EM data acquisition was conducted at the cryo-EM facility in the Center for Structural Biology, NCI-Frederick, using a Talos Arctica G2 (Thermo Fisher) equipped with a Gatan K3 direct detector and energy filter, operated at 200 keV. Data were collected in super-resolution mode at a nominal magnification of 100,000×, corresponding to 0.405 Å/pixel. 50 frames per movie were acquired for a total dose of approximately 50 elections/Å². Data collection was managed with EPU software (Thermo Fisher), setting defocus values ranging from −0.8 to −2.5 μm. Cryo-EM data analysis was performed using Cryosparc 3.3[95]. Movies were imported, patch-motion and patch-CTF corrected. Movies were binned to the physical pixel size in the patch-motion step. Micrographs with CTF resolution >5 Å or with visible bad ice were excluded. An initial subset of particles was picked using blob picker and used to train a Topaz[112,113] model that was used to pick particles in the entire data set. Particles were curated using multiple rounds of 2D classifications, after which duplicated particles were removed. An initial 3D classification with Ab-initio and heterogeneous refinement[114] identified distinct volumes corresponding to unbound LRRK2, LRRK2:14-3-3₂ complex, and LRRK2 dimer. Heterogeneous refinement was iterated until no new conformers appeared.

Further non-uniform refinement[114] of the LRRK2:14-3-3₂ class achieved a map with 3.96 Å resolution, as determined by the gold standard FSC. The structural dynamics of the complex were analyzed using 3D variability analysis[115] with a 7 Å filter. Local refinement of the Roc-COR:14-3-3₂ portion of the map improved the resolution to 3.87 Å. Graphical summary of the cryo-EM data processing is presented in Supplementary Fig. 5c. Cryo-EM reconstruction statistics are detailed in Supplementary Table 2. The global refined map was additionally post-processed using deepEMhancer v0.13[116]. The refined maps were deposited in the EMDB database. The LRRK2:14-3-3₂ complex model was built by the rigid-body fitting of individual monomeric proteins, LRRK2 (PDB 7LHW, modified to include T1647S and T2397M mutations as in WT sequence and have residues 540–703 deleted) and 14-3-3γ (PDB 2B05), into the global refined map. Each protomer of the 14-3-3 dimer was fitted individually, and the LRRK2 phospho-binding sites on each 14-3-3 protomer were built manually. Fitting the models into their respective maps was initially done using UCSF Chimera[117]. Manual

adjustments were performed in Coot[118] using the global and local refined maps, followed by iterative rounds of real-space refinement in Phenix[119] and manual fitting in Coot[118]. Model validation was performed based on statistics from Ramachandran plots and MolProbity scores from Phenix and Coot[120,121]. Statistics for the final refinements are presented in Supplementary Table 2. Figures were prepared using UCSF ChimeraX[122] with structural analyses for structure deviations and electrostatic potential of surfaces performed using its MatchMaker and Coulombic Surface tools, respectively. Final coordinates were deposited in the PDB database and the maps in the EMDB.

### Differential scanning fluorometry

Gel filtration LRRK2 and LRRK2/14-3-3 complex samples, at a concentration of ~0.1 mg/ml, were subjected to thermal stability measurements using a Prometheus NT.48 nano-DSF instrument (NanoTemper) using nano-DSF glass capillaries. Thermal unfolding of the proteins was measured by heating the samples at a rate of 1 °C/min. Protein melting temperatures were calculated by analyzing the first derivative of the ratio of tryptophan fluorescence intensities at 330 nm and 350 nm.

### Mass spectrometry data acquisition and analysis

LRRK2:14-3-3₂ samples were prepared by reducing with 3 mM Tris(2-carboxyethyl)phosphine (TCEP) hydrochloride at room temperature for 1 h, followed by alkylation with 5 mM N-Ethylmaleimide for 10 min. Proteins were then digested with trypsin (Trypsin Gold, Mass Spectrometry Grade, Promega) using a 1:20 enzyme to sample ratio (w/w) at 37 °C for 18 h. The digested samples were desalted using a μElution HLB plate (Waters).

Mass spectrometry data acquisition was performed on a system where an Ultimate 3000 HPLC (Thermo Scientific) was coupled to an Orbitrap Lumos mass spectrometer (Thermo Scientific) via an Easy-Spray ion source (Thermo Scientific). Peptides were separated on an ES902 Easy-Spray column (Thermo Scientific). The composition of mobile phases A and B was 0.1% formic acid in HPLC water, and 0.1% formic acid in HPLC acetonitrile, respectively. The mobile B amount was increased from 3% to 20% in 63 min at a flow rate of 300 nL/min. The Thermo Scientific Orbitrap Lumos mass spectrometer was operated in data-dependent mode. MS1 scans were performed in Orbitrap with a resolution of 120 K at 200 m/z and a mass range of 375–1500 m/z. MS2 scans were conducted in an ion trap. Higher energy collisional dissociation (HCD) method was used for MS2 fragmentation with normalized energy at 32%.

Database search was performed with Proteome Discoverer 2.4 software using the Mascot search engine, against a house-built database containing the sequences of interest and Sprot Human database. The mass tolerances for precursor and fragment were set to 5 ppm and 0.6 Da, respectively. Up to 2 missed cleavages were allowed for data obtained from trypsin digestion. NEM on cysteines was set as a fixed modification. Variable modifications include Oxidation (M), Met-loss (Protein N-term), Acetyl (Protein N-term), and Phosphorylation (STY). Peptides matched with phosphorylation modification were manually curated.

### Co-immunoprecipitation and western blot analysis of LRRK2 and 14-3-3

HEK293FT cells transfected with WT or mutant LRRK2 plasmids were lysed in lysis buffer (20 mM Tris-HCl pH 7.5, 150 mM NaCl, 1 mM EDTA, 0.3% Triton X-100, 10% Glycerol, 1× Halt phosphatase inhibitor cocktail from Thermo Scientific and protease inhibitor cocktail from Roche) for 30 min on ice. The lysates were centrifuged at 4 °C for 10 min at 20,000 × g, and the supernatant was further cleared by incubation with Easy View Protein G agarose beads (Sigma–Aldrich) for 30 min at 4 °C. After removing the agarose beads by centrifugation, the supernatants were incubated with FLAG-M2 affinity resin (Sigma) for 1 h at

4 °C on a rotator. The beads were washed four times with wash buffer (20 mM Tris-HCl pH 7.5, 150 mM NaCl, 1 mM EDTA, 0.1% Triton X-100, 10% Glycerol) and eluted in elution buffer (25 mM Tris-HCl, pH 7.5, 5 mM beta-glycerophosphate, 2 mM dithiothreitol DTT, 0.1 mM $Na_3VO_4$, 10 mM MgCl2, 150 mM NaCl, 0.02% Triton and 150 ng/µl of 3X-FLAG peptide (Sigma–Aldrich)) by shaking for 30 min at 4 °C. Each co-immunoprecipitation was quantified as a ratio between two immuno-precipitated proteins, endogenous 14-3-3 and LRRK2. Proteins were resolved on 4–20% Criterion TGX pre-cast gels (Bio-Rad) in SDS/Tris-glycine running buffer and transferred to membranes using the semi-dry trans-Blot Turbo transfer system (Bio-Rwad). Membranes were blocked with Odyssey Blocking Buffer (Li-Cor cat. 927-40000) and then incubated overnight at 4 °C with the primary antibodies for anti-LRRK2 (Abcam cat. ab133474, dilution 1:2000) and anti-pan-14-3-3 (Santa Cruz cat. sc-133233, dilution 1:2000) and anti-Cyclophilin B (Abcam cat. ab16045, dilution 1:2000). The membranes were washed in TBS-T three times for 5 min followed by incubation for 1 h at room temperature with fluorescently conjugated goat anti-mouse or rabbit IRDye 680 or 800 antibodies (Li-Cor). The blots were washed in TBS-T three times for 5 min at room temperature and scanned on an ODYSSEY® CLx (Li-Cor). Quantitation of western blots was performed using ImageStudio (Li-Cor). LRRK2 pS935 and pS910 levels were normalized to total LRRK2 levels, and pRab10 levels were normalized to total Rab10 levels.

## Microscale thermophoresis (MST) for LRRK2–14-3-3 binding affinity measurement

Binding affinity between LRRK2 and 14-3-3 proteins was determined using microscale thermophoresis (MST). LRRK2 was labeled using the RED-NHS 2nd generation kit and used at a constant concentration of 40 nM. The 14-3-3 protein was prepared at an initial stock concentration of 485 µM, followed by serial 1:2 dilutions across 16 tubes to create a range of 14-3-3 concentrations. Samples were mixed by combining labeled LRRK2 with each dilution of 14-3-3 and incubated to allow binding equilibrium at 25 °C for 1 h. When applicable, MLi-2 and Rebastinib inhibitors were added at 2uM. MST measurements were performed using a Monolith instrument (NanoTemper Technologies) at medium MST power and an infrared laser power (IR-laser) of 20%. Each binding curve was performed in duplicate using three independent protein preparations. Data were analyzed using the instrument's analysis software, and binding affinities were determined by fitting the binding curves to a standard 1:1 binding model. The 95% confidence interval (CI) of the calculated dissociation constants ($K_D$) is reported.

## Reporting summary

Further information on research design is available in the Nature Portfolio Reporting Summary linked to this article.

## Data availability

The cryo-EM density maps were deposited in the Electron Microscopy Data Bank (EMDB), with accession code EMD-45609 for the LRRK2:14-3-3$_2$ global refinement map. The corresponding atomic model for the LRRK2:14-3-3$_2$ complex was deposited in the Protein Data Bank (PDB) under accession code 9CI3. PDB codes of previously published structures used in this study are 8FO9, 7LHW, and 2B05. Additional data generated during this study, including results from kinase assays, inhibition assays, co immunoprecipitation, mass spectroscopy, SEC-MALS, and differential scanning fluorimetry melting temperature data, are provided in the Source Data files. Source data are provided with this paper.

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

## Acknowledgements

This research was supported by Federal funds from the National Cancer Institute, National Institutes of Health, under project number ZIA BC 011744 (P.Z.), and was additionally supported in part by the Intramural Research Program of the National Institute on Aging, National Institutes of Health. This work utilized the Center for Structural Biology cryo-EM facility, NCI at Frederick, and we would like to thank Dr. Dan Shi for his assistance with cryo-EM data collection. We also thank Dr. Yan Li from the protein/peptide sequencing facility at the National Institute of Diabetes and Digestive and Kidney Diseases for mass spectrometry data collection. We thank Dr. Sergey G. Tarasov and Marzena Dyba for assistance with collecting the differential scanning fluorometry and mass photometry data in the Biophysics Resource at the Center for Structural Biology, NCI at Frederick. Additionally, this study utilized the computational resources from the Frederick Research Computing Environment cluster.

## Author contributions

J.A.M.F. and P.Z. conceptualized the project. J.A.M.F. collected and processed the cryo-EM data with input from R.M. and P.Z. J.A.M.F., N.L. and A.A.C. prepared all recombinant protein samples. J.A.M.F. undertook the biochemical/biophysical characterization of protein samples. J.A.M.F. and R.M. performed the MALS experiment. A.B. performed the in vitro cell assays. M.R.C. and P.Z. supervised the work. J.A.M.F. and P.Z. wrote the first draft of the manuscript, and all authors approved the final version of the manuscript.

## Funding

## Competing interests

The authors declare no competing interests.
