## [Transparent Peer Review file · Nature Communications]

14-3-3 binding maintains the Parkinson's associated kinase LRRK2 in an inactive state

Corresponding Author: Dr Ping Zhang

Version 0:

Reviewer comments:

Reviewer #1

(Remarks to the Author)

In their manuscript the authors present a long-awaited structure of LRRK2 in complex with a 14-3-3 (γ) protein, which acts as a natural regulator of LRRK2 activity, using single particle cryo-EM. Kinase-activating mutations in LRRK2 are a leading cause of Parkinson's disease (PD). Additionally, binding of 14-3-3 proteins to LRRK2 is impaired by certain PD mutations, leading to increased LRRK2 activity. This puts forward the LRRK2-14-3-3 interface as a potential target for drug development. Previously, a crystal structure of a 14-3-3 protein in complex with phosphorylated LRRK2 peptides was published (Stevens *et al*, *Biochem J.*, 2017), but structures revealing the entire interaction surface between both proteins was lacking. Although the resolution of the structure is relatively low ($\pm 4\text{\AA}$), it provides important new information regarding previously unidentified secondary interfaces between the 14-3-3 and the LRRK2 CORA and CORB domains, next to the anticipated interactions with the pS910 and pS935 phosphorylation sites. The authors propose that these interactions stabilize LRRK2 in an inactive conformation, by keeping the LRR domain in an auto-inhibiting conformation. However, as further elaborated on below, we have concerns about the validity of this latter mechanism.

While in general we believe the results described in the paper are interesting and could present a significant step forward in the understanding of the regulation of LRRK2, we do have important concerns and remarks, as detailed below.

Major Remarks:

- Previous reports showed that LRRK2 peptides phosphorylated on position S1444 strongly interact with 14-3-3 γ (Manschuetus *et al*. *Front Neurosci.* 2020). In the current structure this interaction is not observed, which is explained by the observation that the region surrounding S1444 is occluded by LRRK2 intramolecular interactions. Fig. S9B presents a list of LRRK2 phosphosites retrieved by MS. However, this list only includes phosphosites up to position 976. Does this mean that there are no phosphosites retrieved C-terminal from position 976? Can the authors comment on whether S1444 is found phosphorylated or not? If S1444 is really occluded by intramolecular interactions, probably also no phosphorylation in this position should be found.

- A mutational approach in combination with co-IP is used to assess the contribution of COR residues to the interaction with 14-3-3. While this method provides some qualitative insights regarding the interaction, it does not provide any quantitative measure of the contribution of these residues. Considering that LRRK2 and 14-3-3 are available in a pure form, it would be advisable to use a more quantitative method to assess the interaction (e.g. SPR, MST, BLI, ...).

- We do not understand the author's interpretation / explanation regarding the effect of the 14-3-3 protein on the LRRK2 monomer/dimer equilibrium (Lines 210-232). On the one hand the authors observe that the 14-3-3 interface partially overlaps with the LRRK2 COR:COR dimerization interface, while on the other hand they observe via MALS that 14-3-3 does not disrupt the LRRK2 dimer and even binds to the LRRK2 dimer. First, it should be described how this dimer was induced and homogenized. Next, the authors propose from these observations that "the overlap in the interface may result in reduced dimerization affinity, so 14-3-3 binding may competitively inhibit LRRK2 dimerization". However, this explanation does not make sense in our opinion. If 14-3-3 acts as a competitive (dose-dependent) inhibitor of LRRK2 dimerization it should not bind to a LRRK2 dimer. To resolve this issue we propose that the authors conduct a mass photometry experiment at different LRRK2 concentrations to assess the LRRK2 monomer-dimer equilibrium, and subsequently investigate the effect of 14-3-3 on this equilibrium. Alternatively, a negative stain EM experiment can be performed where the monomers and dimers are

identified on the micrographs at increasing concentrations of 14-3-3.

- Figure 2 describes “detailed interactions” of the binding interface. However, the interpretation of the LRRK2 regions containing pS910 and pS935 is based on very poor density. While it is possible that those are LRRK2 regions (note that the missing C-terminal of 14-3-3 is very close), the density in this region does not allow for identification of any residues on the LRRK2 side, nor to position any side chains on the 14-3-3 side. Therefore, the primary interactions described on figure 2b cannot be inferred from the structural data. It is shown that S910A and S935A mutations on LRRK2 completely abolish interactions on Co-IP assays, if mutants of 14-3-3 amino acids expected to interact with the S910/S935 phosphorylation site are tested and would show the same effect it would be a step towards the validation of this 14-3-3 region as the interacting part of those phosphorylation sites.

- Our major concern with the interpretation of the structure concerns the proposed mechanism of 14-3-3-mediated LRRK2 inhibition. The authors propose a scenario where LRRK2 binding to pS910 / pS935 and CORA/CORB regions keeps the LRR domain in an auto-inhibitory conformation, with the LRR wrapping around the kinase active site. However, as the authors point out, the 14-3-3 does not make any direct interactions with the LRR. Moreover, the entire region between amino acid 942 and amino acid 983 (57Å), which links the 14-3-3 binding peptide to the LRR, is flexible and disordered and can hence not be observed in the structure. It is very hard to imagine how this flexible region can provide a mechanism and sufficient rigidity to link 14-3-3 binding to a fixation of the LRR conformation.

- Related to the remark above. As described in the paper, it is known that Type 1 inhibitors inhibit LRRK2 by competing with ATP binding, while inducing an “active LRRK2” conformation where the LRR domain is released from the kinase domain. As the authors point out themselves (line 306) this infers that binding of Type 1 inhibitors would disrupt 14-3-3 binding. Nevertheless, the authors do not test the effect of Type 1 inhibitors on 14-3-3 binding to LRRK2. This is a relatively straightforward experiments to perform. The authors should measure the affinity of 14-3-3 for LRRK2 (using any of the quantitative methods described above) in presence of increasing concentration of a Type1 LRRK2 inhibitor.

- Figure 4C shows representative images of the Western blot of the kinase experiment. Here, pRab10 levels are clearly decreasing as 14-3-3 concentration is increased. However, the amount of total LRRK2 seems to be also decreasing in the same trend as the pRab10, allowing for the alternate interpretation that decreased pRab10 might be due to decreased LRRK2 concentration in the experiment. It is therefore required to normalize the pRab10 levels by the amount of LRRK2 detected on the membrane.

Minor (textual) remarks:

- Line 58. “... acting as dimeric scaffolds modulate a ...” should be “... acting as dimeric scaffolds that modulate a ...”

- Line 319 states that human mutation R1628P is located at the COR-A/14-3-3 interface, however, this is not supported by the model they provide, there are not any regions of 14-3-3 at interacting distance of R1628.

Reviewer #2

(Remarks to the Author)

Reviewer #3

(Remarks to the Author)

The manuscript by Martinez Fiesco and colleagues makes a valuable contribution to the understanding of LRRK2 regulation through its interaction with 14-3-3 proteins. Utilizing a cryo-EM structure at sub-4 Å resolution, the authors provide robust structural data elucidating how 14-3-3 binding at phosphorylation sites pS910 and pS935 stabilizes LRRK2 in an inactive monomeric state. This mechanistic explanation builds on the critical role of the COR domain in LRRK2 oligomerisation and underscores the functional significance of 14-3-3 interactions in maintaining the inactive conformation. By stabilizing the LRR domain, 14-3-3 binding prevents LRRK2 dimerisation and subsequent activation. The authors also present evidence for secondary contact sites for 14-3-3 within the COR domain, beyond the primary phospho-motif binding sites at pS910 and pS935. Their findings provide support for the phosphorylation site protection mechanism reported in previous studies.

While this work offers promising insights into LRRK2 regulation and lays a foundation for further investigations, additional biochemical experiments would significantly strengthen the hypotheses presented.

Detailed Critique

1. Potential structural bias A potential limitation of the study arises from the use of chemical crosslinking during sample preparation, which could stabilise non-native conformations. While crosslinking is often invaluable for structural studies, the authors should discuss the possibility of introducing structural bias. Moreover, the unexpectedly high number of particles selected for reconstruction raises questions regarding dataset heterogeneity. Clarifying the proportion of particles

representing the presented structure would improve transparency. The authors are encouraged to provide more detailed information on their data processing workflow, including particle selection criteria.

2. Phosphorylation site protection. The manuscript proposes and validates secondary contact sites for 14-3-3 on LRRK2 beyond the primary phospho-motif binding sites at pS910 and pS935. The data suggest that the dephosphorylation of these primary sites significantly reduces the LRRK2:14-3-3 interaction, consistent with previous findings. Using LRRK2 and different epitope variants expressed in HEK293 cells, the authors conclude that phosphorylation levels at pS910 and pS935 depend on the shielding effect of 14-3-3, which protects these sites from phosphatase-mediated dephosphorylation. To strengthen this conclusion, aligning co-immunoprecipitation (co-IP) data with measurements of pS935 phosphorylation levels in LRRK2 surface variants at 14-3-3 contact sites (e.g., within the CORA and CORB domains) and in cells would provide more direct evidence for the proposed protection mechanism.

3. Enhanced affinity by a COR interface vs. steric hindrance. The distinction between the enhancement of LRRK2:14-3-3 interactions through secondary epitopes in the COR domain and steric hindrance caused by CORA-to-CORB orientation changes presents a complex interpretation.

Evidence from enhanced binding through hydrophobic residue introduction at contact sites and mutational analysis of the 14-3-3 counterpart supports the positive contribution of these secondary epitopes to binding strength. However, as the conclusions rely primarily on co-IP experiments, direct determination of binding constants would be more quantitative therefore providing stronger evidence.

The authors hypothesise that Type I inhibitors induce a CORA-to-CORB rotation incompatible with 14-3-3 binding, leading to dephosphorylation of pS910 and pS935. To test this hypothesis while preserving pS910/pS935 phosphorylation levels, the authors could perform co-IP experiments using wild-type LRRK2 treated *in vitro* with either Type I or Type II inhibitors. Type I inhibitors stabilise the active conformation of LRRK2, causing significant dephosphorylation of pS910 and pS935, whereas Type II inhibitors stabilise the inactive kinase domain conformation. Although direct binding constant measurements would be ideal, co-IP experiments could sufficiently explore these steric and conformational hypotheses given the potential technical challenges.

Reviewer #4

(Remarks to the Author)

That 14-3-3 proteins bind LRRK2 via phosphorylated Ser910 and Ser935 residues was amongst the first biochemical discoveries of LRRK2, but this important interaction has still not fully been understood. This paper provides a number of new and noteworthy insights, including identifying secondary interaction sites in the COR domain and structural insight into the confusing scenario of LRRK2 kinase inhibitors reduce 14-3-3 binding. These findings will be important to the field. For me the experimental work was well performed and rigorous. I was confused about some of the conclusions and interpretations of the outcomes for clinical translation. But overall this is a good study. Point by point comments are below.

This sentence does not seem to make sense – “14-3-3 are regulatory proteins ubiquitously expressed and abundantly present in cells, acting as dimeric scaffolds modulate a broad spectrum of client proteins through various mechanisms.”

“R1441C/G/H, Y1699C, and I2020T, have diminished 14-3-3 interaction, which 60 is correlated with increased kinase activity”. Does 14-3-3 interaction actually correlate with increased activity? Or maybe just associated? Would also be good to clarify what is meant by diminished.

“Additionally, studies on PD rodent models and analyses of postmortem PD brains show reduced LRRK2 and 14-3-3 interactions, associated with increased kinase activity in sporadic PD” This comment on studies is supported by only one reference (ie study). And I'm not sure if this has been replicated?

“Collectively, these results indicate that 14-3-3 binding is inhibitory for LRRK2 activity.” But G2019S LRRK2 binds 14-3-3 and has increased activity? Is this statement too general?

“To study the interaction between LRRK2 and 14-3-3 proteins, we expressed and purified both proteins separately, utilizing the monomeric form of LRRK2 and 14-3-3 gamma (γ)” While the authors provide a justification for using 14-3-3 gamma perhaps they should be careful in extrapolating this to all 14-3-3 proteins?

“Based on our findings, we proposed that the use of S910/S935 phosphorylation levels may need to be tailored to specific LRRK2 mutations and idiopathic PD conditions that promote the active conformation of LRRK2 and impair 14-3-3 binding.” I'm not quite sure I understand this conclusion. How do you determine iPD conditions that promote an active conformation of LRRK2?

“Additionally, the use of S910/S935 phosphorylation levels as biomarkers may need to be tailored to assess therapeutic responses to inhibitors that trap LRRK2 in an active conformation, thereby reducing 14-3-3 binding.” This is currently what is being done and so your studies are supporting what is currently being done?

“Our results provide a structural basis for understanding key aspects of PD biomarkers, offering a targeted approach to patient stratification in clinical settings.” Again just trying to understand the translational relevance of the study. Here you saying for example it might be possible to screen iPD patients for reduced LRRK2 Ser910/Ser935 phosphorylation and this would stratify patients with LRRK2 trapped in an active conformation? Just trying to visualise how and why this would be

done. Is it possible to further explain?

“Based on our findings, we also propose developing LRRK2 and 14-3-3 protein-protein interaction stabilizers, termed 'glue' molecules, specifically designed to enhance the interaction between LRRK2 and 14-3-3”. This seems a bit speculative and not really within the context of the rest of the paper.

“Given LRRK2's significant role in neurodegenerative disorders, our work highlights the key aspects of PD biomarkers and the therapeutic potential of modulating protein-protein interactions and offers a promising strategy to mitigate the detrimental effects of LRRK2 in PD and related diseases.” Again this seems a bit of an overinterpretation as no data was provided to show therapeutic potential.

Version 1:

Reviewer comments:

Reviewer #1

(Remarks to the Author)

In their revised manuscript, the author have satisfactorily addressed the large majority of our previous concerns. The additional experiments that they performed now better support their claims, and the clarity of the manuscript is improved. In the present form we support publication of the manuscript in Nature Communications.

As a final minor remark we would, however, suggest to remove the new Supplementary Fig. 12. This figure does not really provide any added value to the manuscript. While the authors suggest, based on these gels, that the expression and stability of the 14-3-3 mutants is affected, the experiment does not show a purification attempt in the same manner as it was done for the wild type protein, and therefore there is also no direct comparison possible between the left and the right panels of the gel. Alternatively, a sample of the cell lysate for the wild type protein, obtained under the same conditions as the mutants, should be added to the left gel to allow direct comparison.

Reviewer #2

(Remarks to the Author)

Reviewer #3

(Remarks to the Author)

The additional experiments and clarifications have fully addressed my previous concerns. In particular, the inhibitor experiments added in the revised manuscript significantly strengthen major conclusions drawn from the EM structures. I support the publication of this manuscript, pending minor revisions.

Minor point:

Line 194ff (newly added data): In the revised manuscript, the authors state:

“Our mass spectrometry data detected no phosphorylation sites beyond residue 976 (Supplementary Fig. 9b), indicating that phosphorylation of LRRK2 in the LRRK2:14-3-3₂ complex is restricted to the N-terminal half of the protein.”

While the data may be accurate, I find the conclusion too strong given the evidence presented. This statement presumes adequate mass spectrometric coverage of the C-terminal region, including the S2524 site. To substantiate this interpretation, the authors should provide a sequence coverage map, specifically showing whether the C-terminal peptide containing S2524 was detected in their MS analysis. If this region was not confidently covered, the conclusion should be tempered accordingly.

Reviewer #4

(Remarks to the Author)

The applicants have addressed all my comments sufficiently and improved the paper.

Response to the reviewer's comments:

We sincerely appreciate the reviewers for their supportive comments and constructive suggestions. In response, we have addressed all concerns point by point, (responses shown in blue) and made substantial revisions to the manuscript. Specifically, we now include new quantitative data on the LRRK2/14-3-3 interaction using microscale thermophoresis (MST), analysis of additional 14-3-3 mutants, and assessment of the impact of both Type I and Type II inhibitors on LRRK2–14-3-3 binding *in vitro*. We also have incorporated new cell-based assays evaluating inhibitor-induced changes in LRRK2 pS910 and pS935 phosphorylation. We have revised the text to incorporate these new findings and to directly address the reviewers' specific comments.

Reviewer #1 (Remarks to the Author):

In their manuscript the authors present a long-awaited structure of LRRK2 in complex with a 14-3-3 (γ) protein, which acts as a natural regulator of LRRK2 activity, using single particle cryo-EM. Kinase-activating mutations in LRRK2 are a leading cause of Parkinson's disease (PD). Additionally, binding of 14-3-3 proteins to LRRK2 is impaired by certain PD mutations, leading to increased LRRK2 activity. This puts forward the LRRK2-14-3-3 interface as a potential target for drug development. Previously, a crystal structure of a 14-3-3 protein in complex with phosphorylated LRRK2 peptides was published (*Stevens et al, Biochem J., 2017*), but structures revealing the entire interaction surface between both proteins was lacking. Although the resolution of the structure is relatively low ($\pm 4\text{\AA}$), it provides important new information regarding previously unidentified secondary interfaces between the 14-3-3 and the LRRK2 CORA and CORB domains, next to the anticipated interactions with the pS910 and pS935 phosphorylation sites. The authors propose that these interactions stabilize LRRK2 in an inactive conformation, by keeping the LRR domain in an auto-inhibiting conformation. However, as further elaborated on below, we have concerns about the validity of this latter mechanism.

While in general we believe the results described in the paper are interesting and could present a significant step forward in the understanding of the regulation of LRRK2, we do have important concerns and remarks, as detailed below.

Response:

We sincerely thank the reviewer for their positive and thoughtful comments on our work. We appreciate the recognition of the significance of the LRRK2:14-3-3₂ autoinhibitory complex structure, and its potential to advance our understanding of LRRK2 regulation. We also value the reviewer's concerns regarding specific aspects of our proposed mechanism, which we address in detail in the responses below.

Major Remarks:

- Previous reports showed that LRRK2 peptides phosphorylated on position S1444 strongly interact with 14-3-3γ (*Manschuetus et al. Front Neurosci. 2020*). In the current structure this interaction is not observed, which is explained by the observation that the region surrounding S1444 is occluded by LRRK2 intramolecular interactions. Fig. S9B presents a list of LRRK2 phosphosites retrieved by MS. However, this list only includes phosphosites up to position 976. Does this mean that there are no phosphosites retrieved C-terminal from position 976? Can the authors comment on whether S1444 is found phosphorylated or not? If S1444 is really occluded by intramolecular interactions, probably also no phosphorylation in this position should be found.

Response:

We thank the reviewer for raising this important point regarding phosphorylation at S1444 and other C-terminal sites. We would like to clarify the phosphosite mapping presented in Supplementary Fig. 9b was conducted on the purified LRRK2:14-3-3₂ complex, which consists of monomeric, kinase-inactive LRRK2 bound to a 14-3-3 dimer. Prior to complex formation, gel filtration experiment was used to isolate the monomeric form of LRRK2, thereby separating it from dimeric and higher-order assemblies. The phosphorylation profile we obtained thus reflects this specific monomeric population in the complex with 14-3-3.

All phosphorylated sites identified in this sample are listed in Supplementary Fig. 9b. Notably, no phosphorylation was detected beyond residue 976, and we did not observe phosphorylation at S1444 or elsewhere in the C-terminal region. This observation is consistent with our structural finding that the region surrounding S1444 appears occluded by intramolecular interactions (Supplementary Fig. 10b), which may hinder both its phosphorylation and accessibility for 14-3-3 binding in the context of this complex.

To improve clarity, we have revised the main text and the caption of Supplementary Fig. 9b to better describe the composition of the sample used for mass spectrometry.

- A mutational approach in combination with co-IP is used to assess the contribution of COR residues to the interaction with 14-3-3. While this method provides some qualitative insights regarding the interaction, it does not provide any quantitative measure of the contribution of these residues. Considering that LRRK2 and 14-3-3 are available in a pure form, it would be advisable to use a more quantitative method to assess the interaction (e.g. SPR, MST, BLI, ...).

Response:

We greatly appreciate the reviewer's suggestion to apply a quantitative method for assessing the LRRK2/14-3-3 interaction. In response, we have now performed binding affinity measurements using microscale thermophoresis (MST), and the resulting

binding data have now been incorporated into the revised manuscript (Fig. 2e, Supplementary Fig. 11).

Our MST experiments show that wild-type LRRK2 and 14-3-3 γ interact with a dissociation constant (K_D) of approximately 200 nM (Fig. 2e). Furthermore, mutations introduced at key residues within the LRRK2-COR: 14-3-3 γ interface, on either LRRK2 or 14-3-3, significantly reduced the binding affinity, resulting in an approximately 2-3-fold increase in the K_D values (Fig. 2e, Supplementary Fig. 11). These results highlight the functional contribution of the COR:14-3-3 γ secondary interface to complex formation.

We would like to emphasize that while the overall binding between LRRK2 and 14-3-3 remains strong, primarily driven by the interactions with phosphorylated S910 and S935 motifs, the establishment of the COR:14-3-3 secondary interface plays a more critical role in the inhibitory effect of 14-3-3 on LRRK2 kinase activity. This suggests that both interfaces are required for full regulatory function, even if the overall binding affinity is only moderately affected by mutations at the secondary interface.

- We do not understand the author's interpretation / explanation regarding the effect of the 14-3-3 protein on the LRRK2 monomer/dimer equilibrium (Lines 210-232). On the one hand the authors observe that the 14-3-3 interface partially overlaps with the LRRK2 COR:COR dimerization interface, while on the other hand they observe via MALS that 14-3-3 does not disrupt the LRRK2 dimer and even binds to the LRRK2 dimer. First, it should be described how this dimer was induced and homogenized. Next, the authors propose from these observations that "the overlap in the interface may result in reduced dimerization affinity, so 14-3-3 binding may competitively inhibit LRRK2 dimerization". However, this explanation does not make sense in our opinion. If 14-3-3 acts as a competitive (dose-dependent) inhibitor of LRRK2 dimerization it should not bind to a LRRK2 dimer. To resolve this issue we propose that the authors conduct a mass photometry experiment at different LRRK2 concentrations to assess the LRRK2 monomer-dimer equilibrium, and subsequently investigate the effect of 14-3-3 on this equilibrium.

Alternatively, a negative stain EM experiment can be performed where the monomers and dimers are identified on the micrographs at increasing concentrations of 14-3-3.

Response:

We thank the reviewer for highlighting this important point and for noting the need to clarify our interpretation of the LRRK2 monomer/dimer equilibrium in relation to 14-3-3 binding and we have now revised this portion of the text in the manuscript to improve clarity.

To clarify the experimental setup: for our multi-angle light scattering (MALS) experiment, wild-type (WT) LRRK2 containing an N-terminal Flag tag was expressed in mammalian cells and purified using affinity purification followed by size-exclusion chromatography. Mass photometry (MP) analysis of gel filtration fractions revealed that LRRK2 exists in solution as a mixture of high-molecular-weight oligomers, dimers, and monomers

(Extended Data Fig. 1b). For the MALS experiments, we specifically used fractions containing predominantly dimeric LRRK2. The purpose of The MALS experiment was to investigate whether 14-3-3 can bind to LRRK2 homodimers via the primary interaction sites at S910 and S935, which remain accessible in the dimer form (Fig. 3a). We observed that at the highest concentrations of 14-3-3 used, the measured mass (643 KDa) was consistent with a complex containing a single 14-3-3 γ dimer to a single LRRK2 dimer, with no indication of LRRK2 dimer disruption. This suggests that under the experimental conditions used, 14-3-3 γ can bind to the LRRK2 dimer through the primary interface without disrupting the dimer. However, as the reviewer rightly points out, it remains to be determined whether, under physiological conditions, where 14-3-3 proteins are present at concentrations orders of magnitude higher than LRRK2, the partial overlap between the 14-3-3 and at the COR:COR interfaces could eventually lead to destabilization of the LRRK2 dimer. Conversely, the formation of the LRRK2:14-3-3 $_2$ complex, in which the COR interface is occluded, would likely impede LRRK2 dimer formation.

We have revised this portion of the text in the Results section to clarify the experimental design and better align our interpretation with the reviewer's comments.

- Figure 2 describes “detailed interactions” of the binding interface. However, the interpretation of the LRRK2 regions containing pS910 and pS935 is based on very poor density. While it is possible that those are LRRK2 regions (note that the missing C-terminal of 14-3-3 is very close), the density in this region does not allow for identification of any residues on the LRRK2 side, nor to position any side chains on the 14-3-3 side. Therefore, the primary interactions described on figure 2b cannot be inferred from the structural data. It is shown that S910A and S935A mutations on LRRK2 completely abolish interactions on Co-IP assays, if mutants of 14-3-3 amino acids expected to interact with the S910/S935 phosphorylation site are tested and would show the same effect it would be a step towards the validation of this 14-3-3 region as the interacting part of those phosphorylation sites.

Response:

We thank the reviewer for their insightful comment. We agree that the local resolution around the S910 and S935 regions is not atomic and does not alone permit unambiguous placement of side chains. We appreciate the reviewer's close attention to these structural details.

To address this, we have taken a rigorous and integrative approach to assign this primary interaction. In addition to careful model fitting guided by both local and global features of the cryo-EM map, our interpretation is strongly supported by multiple complementary datasets, including previous X-ray crystallographic structural studies of 14-3-3 bound to LRRK2-derived phosphopeptides (Stevens, L. M. et al., 2017. *Biochemical Journal.*), mass spectrometry confirming phosphorylation at S910 and S935, functional co-IP assays, and binding affinity measurements by MST, kinase inhibition experiments, and mutagenesis of LRRK2 (Figs 2, 4, 6 and Supplementary Figs. 8,9). Collectively, these data provide robust evidence that this region mediates the primary interaction between LRRK2 and 14-3-3.

In response to the reviewer's helpful suggestion, we also attempted to validate this interface from the 14-3-3 side by mutating residues within the canonical ligand-binding groove (R57A, K50A, R132A, and the triple mutant R57A/K50A/R132A). Despite extensive efforts, including codon optimization, testing different expression systems, and buffer condition screening, unfortunately these mutants exhibited poor expression and/or instability, which prevented us from carrying out further biochemical characterization (Supplementary Fig. 12). This outcome is not entirely surprising, as mutations in highly conserved and functionally critical residues of scaffolding proteins often lead to cellular toxicity or issues with protein folding and/or stability. While the density in this region alone does not support atomic-level modeling, we believe the convergence of structural, biochemical, and functional data provides compelling support for our assignment. We have therefore retained the structural interpretation shown in Figure 2b and clarified in the figure legend that this assignment is based on integrated evidence. We are grateful to the reviewer for their thoughtful comments, which have helped us to more clearly present the strength of our conclusions.

- Our major concern with the interpretation of the structure concerns the proposed mechanism of 14-3-3-mediated LRRK2 inhibition. The authors propose a scenario where LRRK2 binding to pS910 / pS935 and CORA/CORB regions keeps the LRR domain in an auto-inhibitory conformation, with the LRR wrapping around the kinase active site. However, as the authors point out, the 14-3-3 does not make any direct interactions with the LRR. Moreover, the entire region between amino acid 942 and amino acid 983 (57Å), which links the 14-3-3 binding peptide to the LRR, is flexible and disordered and can hence not be observed in the structure. It is very hard to imagine how this flexible region can provide a mechanism and sufficient rigidity to link 14-3-3 binding to a fixation of the LRR conformation.

Response:

We thank the reviewer for raising this important and thoughtful point. To improve the clarity of our proposed mechanism of 14-3-3-mediated LRRK2 inhibition, we have revised the manuscript with an expanded explanation, now incorporated into the Results section.

We propose that a synergistic interaction between the primary and secondary 14-3-3 binding sites creates a spatial constraint in the LRRK2's domain organization that reinforces the kinase inactive conformation. This constraint does not rely on direct contact between 14-3-3 and the LRR domain but rather arises from the dual anchoring of the LRRK2 loop (containing the phosphorylated pS910/pS935 motifs, upstream of the LRR domain) and the COR-A and COR-B subdomains (downstream of the LRR domain) by the 14-3-3 dimer. Although the intervening loop (residues 943–982) is flexible and unresolved in the cryo-EM map, this spatial arrangement imposes a restriction that limits repositioning of the LRR domain relative to the kinase domain. To further explore this mechanism, we performed 3D variability analysis (3DVA) (Supplementary Fig 14, Supplementary Movie 1), which revealed that the LRR domain

and the 14-3-3 dimer undergo coordinated movements relative to the Roc-COR-Kinase-WD40 core. These synchronized dynamics support the idea that 14-3-3 flexibly tethers the regions flanking the LRR domain, restricting its conformational range and maintaining LRRK2 in an autoinhibited state.

We believe that this combined structural and dynamic evidence provides a plausible and mechanistically supported model for the inhibitory role of 14-3-3 in regulating LRRK2 activity.

- Related to the remark above. As described in the paper, it is known that Type 1 inhibitors inhibit LRRK2 by competing with ATP binding, while inducing an “active LRRK2” conformation where the LRR domain is released from the kinase domain. As the authors point out themselves (line 306) this infers that binding of Type 1 inhibitors would disrupt 14-3-3 binding. Nevertheless, the authors do not test the effect of Type 1 inhibitors on 14-3-3 binding to LRRK2. This is a relatively straightforward experiments to perform. The authors should measure the affinity of 14-3-3 for LRRK2 (using any of the quantitative methods described above) in presence of increasing concentration of a Type1 LRRK2 inhibitor.

Response:

We thank the reviewer for the helpful suggestion. In addition to measuring the binding affinity between LRRK2 and 14-3-3 as described above, we conducted further experiments to assess the effect of LRRK2 conformation on this interaction. Specifically, we measured the affinity between LRRK2 and 14-3-3 in the presence of a saturating concentration of a type I inhibitor (MLi-2), which stabilizes the active conformation, or a type II inhibitor (Rebastinib), which favors the inactive conformation. The results are now included in Fig. 5c and Supplementary Fig. 17.

We observed that the binding affinity between LRRK2 and 14-3-3 was decreased by 2-fold in the presence of type I inhibitor (MLi-2). In contrast, the binding affinity was unaltered in the presence of Type II inhibitor (Rebastinib). These results support our hypothesis that, when LRRK2 adopts an active conformation, its binding to 14-3-3 is impaired, likely due to the conformational rearrangement involving the COR domain that hinders optimal interaction between the two proteins.

- Figure 4C shows representative images of the Western blot of the kinase experiment. Here, pRab10 levels are clearly decreasing as 14-3-3 concentration is increased. However, the amount of total LRRK2 seems to be also decreasing in the same trend as the pRab10, allowing for the alternate interpretation that decreased pRab10 might be due to decreased LRRK2 concentration in the experiment. It is therefore required to normalize the pRab10 levels by the amount of LRRK2 detected on the membrane.

Response:

We thank the reviewer for the suggestion. We have normalized the data to LRRK2 levels, and Figure 4C has been updated accordingly. We would also like to clarify that

the blots shown in the figure are representative of at least six independent measurements used for quantification. In the revised figure, we now present a LRRK2 blot that more accurately reflects the amount of protein used in the kinase assays.

Minor (textual) remarks:

- Line 58. "... acting as dimeric scaffolds modulate a ..." should be "... acting as dimeric scaffolds that modulate a ..."

Response:

We thank the reviewer for pointing this out. We have modified the text accordingly in the revised manuscript.

- Line 319 states that human mutation R1628P is located at the COR-A/14-3-3 interface, however, this is not supported by the model they provide, there are not any regions of 14-3-3 at interacting distance of R1628.

Response:

We thank the reviewer for this important observation. We have revised the text to indicate that the R1628 residue is located near, but not directly at, the COR-A/14-3-3 interface, to more accurately reflect the structural model.

Reviewer #2 (Remarks to the Author):

Response:

We sincerely thank the reviewer for their contribution to the review process of our manuscript.

Reviewer #3 (Remarks to the Author):

The manuscript by Martinez Fiesco and colleagues makes a valuable contribution to the understanding of LRRK2 regulation through its interaction with 14-3-3 proteins. Utilizing

a cryo-EM structure at sub-4 Å resolution, the authors provide robust structural data elucidating how 14-3-3 binding at phosphorylation sites pS910 and pS935 stabilizes LRRK2 in an inactive monomeric state. This mechanistic explanation builds on the critical role of the COR domain in LRRK2 oligomerisation and underscores the functional significance of 14-3-3 interactions in maintaining the inactive conformation. By stabilizing the LRR domain, 14-3-3 binding prevents LRRK2 dimerisation and subsequent activation. The authors also present evidence for secondary contact sites for 14-3-3 within the COR domain, beyond the primary phospho-motif binding sites at pS910 and pS935. Their findings provide support for the phosphorylation site protection mechanism reported in previous studies.

While this work offers promising insights into LRRK2 regulation and lays a foundation for further investigations, additional biochemical experiments would significantly strengthen the hypotheses presented.

Response:

We thank the reviewer for the positive comments and for recognizing the significance of our structural and mechanistic insights into LRRK2 regulation by 14-3-3 proteins. We appreciate the reviewer's thoughtful summary and agree that the findings offer a foundation for further investigations. We have also incorporated additional biochemical experiments, as detailed below, to further strengthen our conclusions.

Detailed Critique

1. Potential structural bias A potential limitation of the study arises from the use of chemical crosslinking during sample preparation, which could stabilise non-native conformations. While crosslinking is often invaluable for structural studies, the authors should discuss the possibility of introducing structural bias. Moreover, the unexpectedly high number of particles selected for reconstruction raises questions regarding dataset heterogeneity. Clarifying the proportion of particles representing the presented structure would improve transparency. The authors are encouraged to provide more detailed information on their data processing workflow, including particle selection criteria.

Response:

We thank the reviewer for raising this important point. We agree that chemical crosslinking has the potential to introduce structural biases by stabilizing non-native conformations. To mitigate this concern, we initially attempted to determine the structure of the LRRK2:14-3-3₂ complex without crosslinking. As shown in Supplementary Fig. 3, the 2D class averages and a low resolution 3D reconstruction obtained from the non-crosslinked sample closely resemble those from the crosslinked sample, supporting the conclusion that the conformation stabilized by crosslinking reflects the native conformation of the complex. Furthermore, the physiological relevance of our cryo-EM model is supported by a comprehensive set of orthogonal experiments, including structure-guided mutagenesis, MST binding assays, in vitro kinase activity assays, and

cellular dephosphorylation studies. Together, these results reinforce the validity of the observed structural state.

We also appreciate the reviewer's concern about the high number of particles used for reconstruction. LRRK2 is a large and highly dynamic multi-domain protein, which contribute to considerable dataset heterogeneity. To address this, we applied stringent *ab initio* and heterogeneous refinement steps to classify and remove poorly aligned particles. The final particle set used for high-resolution refinement represents a well-defined, conformationally homogeneous population. A relatively high number of particles was used to improve map resolution and enable robust 3D variability analyses.

The number of particles used in the final reconstruction, along with the data processing workflow, is provided in Supplementary Fig. 5 and Supplementary Table 2, and described in detail in the Methods section.

2. Phosphorylation site protection. The manuscript proposes and validates secondary contact sites for 14-3-3 on LRRK2 beyond the primary phospho-motif binding sites at pS910 and pS935. The data suggest that the dephosphorylation of these primary sites significantly reduces the LRRK2:14-3-3 interaction, consistent with previous findings.

Using LRRK2 and different epitope variants expressed in HEK293 cells, the authors conclude that phosphorylation levels at pS910 and pS935 depend on the shielding effect of 14-3-3, which protects these sites from phosphatase-mediated dephosphorylation. To strengthen this conclusion, aligning co-immunoprecipitation (co-IP) data with measurements of pS935 phosphorylation levels in LRRK2 surface variants at 14-3-3 contact sites (e.g., within the CORA and CORB domains) and in cells would provide more direct evidence for the proposed protection mechanism.

Response:

We thank the reviewer for this helpful suggestion. To directly test this mechanism, that 14-3-3 shields LRRK2 pS910 and pS935 sites from dephosphorylation, we measured phosphorylation levels at these sites under conditions that weaken the LRRK2:14-3-3₂ interaction. Specifically, we treated HEK293 cells expressing LRRK2 with either the Type I kinase inhibitor MLI-2 or the Type II inhibitor Rebastinib, and additionally evaluated cells expressing LRRK2 mutants that destabilize the COR:14-3-3₂ interface. Consistent with our model, we observed a ~19-fold reduction in pS910 and pS935 levels in cells treated with the type I inhibitor MLI-2, accompanied by a ~7-fold decrease in co-immunoprecipitated 14-3-3 levels (Supplementary Fig. 18a). In contrast, treatment with Rebastinib had no significant effect (Supplementary Fig. 18a). We further examined the impact of interface-disrupting mutations in the COR domain. Mutations L1727A, R1728A, N1730A, E1632A, and L1635A, each located at the COR-A or COR-B subdomains, produced a ~2- to 5-fold reduction in pS935 levels (Supplementary Fig. 18b). These changes were accompanied by a 2- to 13-fold decrease in co-immunoprecipitated 14-3-3 levels (Fig. 2d). These results support the model that 14-3-3 binding to both the primary phospho-motifs and the COR domain not only stabilizes

LRRK2 in its inactive state but also protects pS910 and pS935 from phosphatase-mediated dephosphorylation by sterically shielding these sites.

3. Enhanced affinity by a COR interface vs. steric hindrance. The distinction between the enhancement of LRRK2:14-3-3 interactions through secondary epitopes in the COR domain and steric hindrance caused by CORA-to-CORB orientation changes presents a complex interpretation.

Evidence from enhanced binding through hydrophobic residue introduction at contact sites and mutational analysis of the 14-3-3 counterpart supports the positive contribution of these secondary epitopes to binding strength. However, as the conclusions rely primarily on co-IP experiments, direct determination of binding constants would be more quantitative therefore providing stronger evidence.

The authors hypothesise that Type I inhibitors induce a CORA-to-CORB rotation incompatible with 14-3-3 binding, leading to dephosphorylation of pS910 and pS935. To test this hypothesis while preserving pS910/pS935 phosphorylation levels, the authors could perform co-IP experiments using wild-type LRRK2 treated *in vitro* with either Type I or Type II inhibitors. Type I inhibitors stabilise the active conformation of LRRK2, causing significant dephosphorylation of pS910 and pS935, whereas Type II inhibitors stabilise the inactive kinase domain conformation. Although direct binding constant measurements would be ideal, co-IP experiments could sufficiently explore these steric and conformational hypotheses given the potential technical challenges.

Response:

We thank the reviewer for this valuable suggestion. As recommended, we have performed quantitative binding studies using microscale thermophoresis (MST) to measure the binding affinities between LRRK2 and 14-3-3. These experiments include conditions with wild-type protein, interface mutants at the COR:14-3-3 site, and treatment with the LRRK2 Type I inhibitor MLI-2. This approach complements the co-IP data and provides direct evidence supporting our model that secondary contacts in the COR domain enhance binding affinity, while the conformational rearrangement associated with the active state disrupts 14-3-3 binding.

As this point overlaps with a similar comment raised by Reviewer 1, we have provided a detailed reply in our response to Reviewer 1 to avoid redundancy.

Reviewer #4 (Remarks to the Author):

That 14-3-3 proteins bind LRRK2 via phosphorylated Ser910 and Ser935 residues was amongst the first biochemical discoveries of LRRK2, but this important interaction has still not fully been understood. This paper provides a number of new and noteworthy insights, including identifying secondary interaction sites in the COR domain and structural insight into the confusing scenario of LRRK2 kinase inhibitors reduce 14-3-3 binding. These findings will be important to the field. For me the experimental work was well performed and rigorous. I was confused about some of the conclusions and

interpretations of the outcomes for clinical translation. But overall this is a good study. Point by point comments are below.

Response:

We greatly appreciate the reviewer's positive evaluation of our study. We have carefully considered the reviewer's specific comments and addressed each point in detail below.

This sentence does not seem to make sense – “14-3-3 are regulatory proteins ubiquitously expressed and abundantly present in cells, acting as dimeric scaffolds modulate a broad spectrum of client proteins through various mechanisms.”

Response:

We agree with the reviewer's point and have revised this sentence for clarity. The text now reads: “14-3-3 proteins are ubiquitously expressed and highly abundant in cells. As regulatory proteins, they function as dimeric scaffolds that modulate a broad spectrum of client proteins through various mechanisms.”

“R1441C/G/H, Y1699C, and I2020T, have diminished 14-3-3 interaction, which 60 is correlated with increased kinase activity”. Does 14-3-3 interaction actually correlate with increased activity? Or maybe just associated? Would also be good to clarify what is meant by diminished.

Response:

Thank you for the helpful suggestion. We have revised the sentence for clarity. It now reads: “Several PD-associated mutations such as R1441C/G/H, Y1699C, and I2020T exhibit reduced interaction with 14-3-3 proteins, which is associated with increased kinase activity⁷¹⁻⁷³.”

“Additionally, studies on PD rodent models and analyses of postmortem PD brains show reduced LRRK2 and 14-3-3 interactions, associated with increased kinase activity in sporadic PD” This comment on studies is supported by only one reference (ie study). And I'm not sure if this has been replicated?

Response:

Thank you for this important comment. We acknowledge that the observation regarding reduced LRRK2 and 14-3-3 interactions in sporadic PD is primarily supported by a single study (Di Maio et al., 2018, *Sci. Transl. Med.*). To our knowledge, additional experimental replication in independent cohorts or models is still lacking. We have revised the text to reflect this and now present this point more cautiously. It now reads: "A study of PD rodent models and postmortem PD brain tissue reported reduced LRRK2

and 14-3-3 interactions, which were associated with increased kinase activity in idiopathic PD. "

"Collectively, these results indicate that 14-3-3 binding is inhibitory for LRRK2 activity." But G2019S LRRK2 binds 14-3-3 and has increased activity? Is this statement to general?

Response:

Thank you for pointing this out. We have revised the sentence, based on the reviewer's recommendation, to avoid overgeneralization and to better reflect the complexity of LRRK2 regulation. It now reads:

"Collectively, these findings indicate that 14-3-3 binding can modulate LRRK2 activity, often exerting an inhibitory effect."

"To study the interaction between LRRK2 and 14-3-3 proteins, we expressed and purified both proteins separately, utilizing the monomeric form of LRRK2 and 14-3-3 gamma (γ)" While the authors provide a justification for using 14-3-3 gamma perhaps they should be careful in extrapolating this to all 14-3-3 proteins?

Response:

We thank the reviewer for this important point. While we selected 14-3-3 γ for structural and biochemical studies based on its well-documented interaction with LRRK2, we agree that caution is warranted in generalizing our findings to all other 14-3-3 isoforms. However, several lines of evidence suggest that our observations likely extend to other isoforms. Specifically, mass spectrometry analysis of the LRRK2:14-3-3₂ complex revealed the presence of multiple endogenous 14-3-3 isoforms co-purified with LRRK2 from mammalian cells, despite the use of purified recombinant 14-3-3 γ (Supplementary Fig. 2b). Moreover, sequence alignment of all seven human 14-3-3 isoforms revealed that the residues involved in both primary and secondary interactions with LRRK2 are highly conserved (Supplementary Fig. 7b). Altogether, these findings support the interpretation that the structural and mechanistic principles described here are likely applicable across the 14-3-3 family. We have clarified this point in the revised text.

"Based on our findings, we proposed that the use of S910/S935 phosphorylation levels may need to be tailored to specific LRRK2 mutations and idiopathic PD conditions that promote the active conformation of LRRK2 and impair 14-3-3 binding." I'm not quite sure I understand this conclusion. How do you determine iPD conditions that promote an active conformation of LRRK2?

Response:

We thank the reviewer for pointing this issue out. We agree that the conformational state of LRRK2 in idiopathic PD is not yet clearly defined, and that our original statement may have implied greater certainty than warranted. Our intent was to highlight that the phosphorylation status of pS910/pS935 may be differentially affected in disease contexts where 14-3-3 binding is impaired, whether due to pathogenic mutations or potential regulatory disruptions observed in idiopathic PD. To clarify this point, we have revised the sentence and now it states: “Based on our findings, we propose that the use of pS910/pS935 phosphorylation levels as biomarkers may need to be tailored to specific LRRK2 mutations that promote the active conformation and impair 14-3-3 binding. This approach may also be relevant to idiopathic PD, which has been associated with reduced LRRK2–14-3-3 interactions. While it remains unclear whether LRRK2 adopts an active conformation in idiopathic PD, the observed decrease in 14-3-3 binding and increased susceptibility of pS910/pS935 to dephosphorylation in patient samples suggest that these phosphorylation sites may serve as useful biomarkers for stratifying LRRK2 regulatory states and identifying patients with similar LRRK2 regulatory dysfunctions.”

“Additionally, the use of S910/S935 phosphorylation levels as biomarkers may need to be tailored to assess therapeutic responses to inhibitors that trap LRRK2 in an active conformation, thereby reducing 14-3-3 binding.” This is currently what is being done and so your studies are supporting what is currently being done?

Response:

We thank the reviewer for pointing this out. We have modified the text to better reflect the current clinical context and the contribution of our findings. It now reads: “Additionally, the use of S910/S935 phosphorylation levels as biomarkers may need to be tailored to assess therapeutic responses to inhibitors that trap LRRK2 in an active conformation, thereby reducing 14-3-3 binding. Our findings support the current use of these phosphorylation levels in clinical trials and provide structural and mechanistic insights into how these biomarkers reflect LRRK2's conformational state and interaction with 14-3-3. This understanding strengthens the rationale for using these biomarkers and may guide more precise patient stratification in clinical settings.”

“Our results provide a structural basis for understanding key aspects of PD biomarkers, offering a targeted approach to patient stratification in clinical settings.” Again just trying to understand the translational relevance of the study. Here you saying for example it might be possible to screen iPD patients for reduced LRRK2 ser910/ser935 phosphorylation and this would stratify patients with LRRK2 trapped in an active confirmation? Just trying to visualise how and why this would be done. Is it possible to further explain?

Response:

We thank the reviewer for the thoughtful question. Our findings suggest that reduced LRRK2 Ser910/Ser935 phosphorylation levels could serve as a useful biomarker for identifying idiopathic PD patients with disrupted LRRK2 regulation. While it is not yet established whether LRRK2 in idiopathic PD patients adopts an active conformation, our study shows that reduced Ser910/Ser935 phosphorylation correlates with impaired 14-3-3 binding and increased kinase activity, both hallmarks of LRRK2 pathogenic LRRK2 states. Clinically, stratifying idiopathic PD patients based on reduced Ser910/Ser935 phosphorylation could help identify those with similar regulatory disruptions. Additionally, tracking dynamic changes in these phosphorylation levels could be used to monitor therapeutic responses, particularly to LRRK2-targeted treatments.

We have revised the manuscript to clarify this point and to emphasize the translational potential of our findings, while acknowledging current limitations in defining LRRK2 conformational states in idiopathic PD.

“Based on our findings, we also propose developing LRRK2 and 14-3-3 protein-protein interaction stabilizers, termed 'glue' molecules, specifically designed to enhance the interaction between LRRK2 and 14-3-3”. This seems a bit speculative and not really within the context of the rest of the paper.

Response:

Thank you for pointing this out. We agree that the suggestion to develop LRRK2/14-3-3 interaction 'glue' molecules may appear speculative in the context of our current study. We have revised the text to clarify this is a conceptual extension of our findings, intended to inspire future research rather than being a conclusion directly supported by the data presented here.

“Given LRRK2's significant role in neurodegenerative disorders, our work highlights the key aspects of PD biomarkers and the therapeutic potential of modulating protein-protein interactions and offers a promising strategy to mitigate the detrimental effects of LRRK2 in PD and related diseases.” Again this seems a bit of an overinterpretation as no data was provided to show therapeutic potential.

Response:

Thank you for your feedback. We acknowledge that the statement regarding therapeutic potential may have exceeded the scope of the data presented. Accordingly, we have revised the text to focus more clearly on the structural and mechanistic insights provided by our study. It now reads: “These findings offer valuable insights into the molecular mechanisms underlying LRRK2 regulation and lay a foundation for future therapeutic strategies targeting this pathway. Our work also informs the interpretation of PD biomarkers and supports further exploration of 14-3-3 based regulatory mechanisms in disease contexts.”

Response to the reviewer's comments:

We sincerely thank all reviewers for their thorough evaluation of our manuscript, their constructive suggestions which strengthened the study, and their final support for its publication. The final specific comments raised by Reviewers #1 and #3 have been addressed in the revised manuscript and are detailed below.

Reviewer #1 (Remarks to the Author):

In their revised manuscript, the author have satisfactorily addressed the large majority of our previous concerns. The additional experiments that they performed now better support their claims, and the clarity of the manuscript is improved. In the present form we support publication of the manuscript in Nature Communications. As a final minor remark we would, however, suggest to remove the new Supplementary Fig. 12. This figure does not really provide any added value to the manuscript. While the authors suggest, based on these gels, that the expression and stability of the 14-3-3 mutants is affected, the experiment does not show a purification attempt in the same manner as it was done for the wild type protein, and therefore there is also no direct comparison possible between the left and the right panels of the gel. Alternatively, a sample of the cell lysate for the wild type protein, obtained under the same conditions as the mutants, should be added to the left gel to allow direct comparison.

We thank the reviewer for their overall positive assessment of our manuscript and for supporting its publication. We also appreciate the comment regarding Supplementary Fig. 12. We agree that a direct comparison between the left and right panels is not feasible due to differences in sample preparation. However, we believe that the figure provides useful context, particularly in showing that the 14-3-3 variants harboring mutations in the canonical ligand-binding groove (R57A, K50A, R132A, and the triple mutant) are poorly expressed, likely due to misfolding or instability. This supports our interpretation that these residues are critical for 14-3-3 structural integrity.

The gel of the wild-type protein is included solely as a reference for migration size and typical expression level, not for direct quantitative comparison. To clarify this, we have revised the figure legend to explicitly note that the gels are not directly comparable and that the WT gel is provided as a reference only. It now reads:

“Supplementary Fig. 12. Expression analysis for 14-3-3 mutants. a. Representative small-scale expression trials of 14-3-3 γ mutants harboring individual substitutions at residues R132, R57, and K50, as well as the triple mutant R57A/K50A/R132A, show poor or no detectable expression, suggesting that these mutations impact protein stability. b. A sample of purified WT 14-3-3 γ is shown for reference to indicate the expected migration and purity. Note that the WT sample was obtained through a

standard purification protocol and is not directly comparable to the mutant lysates in panel a. This figure is included to highlight the apparent loss of expression or solubility in the mutants, not for quantitative comparison.”

We hope the reviewer finds this clarification satisfactory, and we respectfully request to retain the figure in the Supplementary Information.

Reviewer #3 (Remarks to the Author):

The additional experiments and clarifications have fully addressed my previous concerns. In particular, the inhibitor experiments added in the revised manuscript significantly strengthen major conclusions drawn from the EM structures. I support the publication of this manuscript, pending minor revisions.

Minor point:

Line 194ff (newly added data): In the revised manuscript, the authors state: “Our mass spectrometry data detected no phosphorylation sites beyond residue 976 (Supplementary Fig. 9b), indicating that phosphorylation of LRRK2 in the LRRK2:14-3-3₂ complex is restricted to the N-terminal half of the protein.”

While the data may be accurate, I find the conclusion too strong given the evidence presented. This statement presumes adequate mass spectrometric coverage of the C-terminal region, including the S2524 site. To substantiate this interpretation, the authors should provide a sequence coverage map, specifically showing whether the C-terminal peptide containing S2524 was detected in their MS analysis. If this region was not confidently covered, the conclusion should be tempered accordingly.

We thank the reviewer for their supportive feedback and helpful suggestion. We agree that, in the absence of a detail sequence coverage map demonstrating confident detection of the C-terminal region, including S2524, the conclusion regarding the absence of phosphorylation should be presented more cautiously.

Accordingly, we have revised the sentence to read:

“Our mass spectrometry data did not identify phosphorylation sites beyond residue 976 (Supplementary Fig. 9b), suggesting that phosphorylation of LRRK2 in the LRRK2:14-3-3₂ complex may be primarily limited to the N-terminal half of the protein; however, further studies with comprehensive sequence coverage will be needed to confirm this observation.”

We hope this modification appropriately addresses the reviewer’s concern.

No other changes were made to the manuscript beyond those noted above. We are grateful for the time and effort of the reviewers and editorial team and look forward to the next steps in the publication process.